# GemDepth: Geometry-Embedded Features for 3D-Consistent Video Depth

Yuecheng Liu [1]   Junda Cheng [1]   Longliang Liu [1 2]   Wenjing Liao [1 2]   Hanrui Cheng [1 2]   Yuzhou Wang [1]
Xin Yang [1 3]

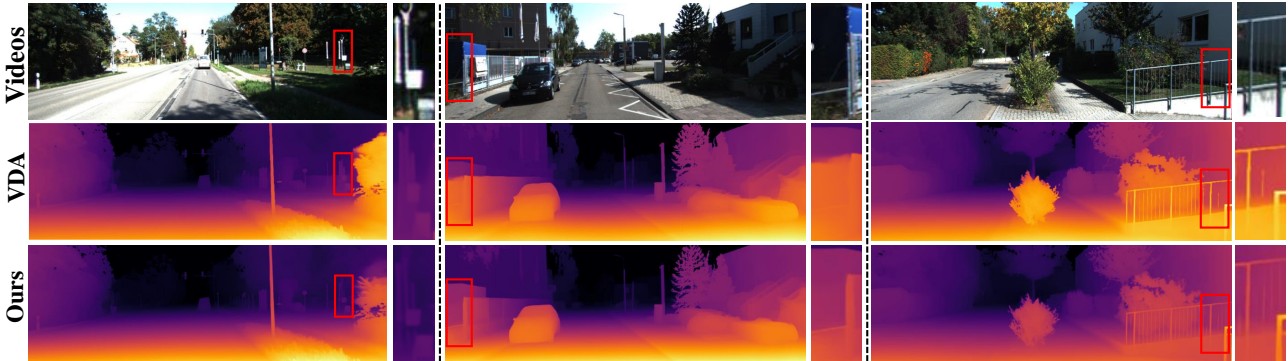

*Figure 1.* **Qualitative comparison on the KITTI (Geiger et al., 2013) dataset.** Compared with the state-of-the-art VideoDepthAnything (VDA) (Chen et al., 2025), our GemDepth-VDA demonstrates superior capability in resolving intricate background details and preserving fine structures. Left: Full-resolution of the video depth sequences; Right: Zoomed-in views.

## Abstract

Video depth estimation extends monocular prediction into the temporal domain to ensure coherence. However, existing methods often suffer from spatial blurring in fine-detail regions and temporal inconsistencies. We argue that current approaches, which primarily rely on temporal smoothing via Transformers, struggle to maintain strict 3D geometric consistency—particularly under rotations or drastic view changes. To address this, we propose GemDepth, a framework built on the insight that an explicit awareness of camera motion and global 3D structure is a prerequisite for 3D consistency. Distinctively, GemDepth introduces a Geometry-Embedding Module (GEM) that predicts inter-frame camera poses to generate implicit geometric embeddings. This injection of motion priors equips the network with intrinsic 3D perception and alignment capabilities. Guided by these geometric cues, our Alternating Spatio-Temporal Transformer (ASTT) captures latent point-level correspondences to si-

multaneously enhance spatial precision for sharp details and enforce rigorous temporal consistency. Furthermore, GemDepth employs a data-efficient training strategy, effectively bridging the gap between high efficiency and robust geometric consistency. As shown in Fig. 2, comprehensive evaluations demonstrate that GemDepth achieves state-of-the-art performance across multiple datasets, particularly in complex dynamic scenarios. The code is publicly available at: https://github.com/Yuecheng919/GemDepth.

## 1. Introduction

Monocular depth estimation is a cornerstone of computer vision, underpinning applications from autonomous driving to augmented reality (Holynski & Kopf, 2018; Liu et al., 2019; Luo et al., 2020; Hu et al., 2021; Hong et al., 2022; Watson et al., 2023; Gu et al., 2025; Cheng et al., 2024; 2025b;a;c). Despite the significant strides of powerful foundation models (Samuel, 1959; Oquab et al., 2023; Yin et al., 2023; Bochkovskii et al., 2024; Yang et al., 2024b; Piccinelli et al., 2024; Ke et al., 2024; Fu et al., 2024), these frameworks are primarily designed for single-frame estimation. By treating frames in isolation, they neglect the temporal dependencies crucial for video coherence. Consequently, this lack of cross-frame awareness often leads to severe

[1]Huazhong University of Science & Technology [2]Carizon [3]Optics Valley Laboratory. Correspondence to: Junda Cheng <jundacheng@hust.edu.cn>.

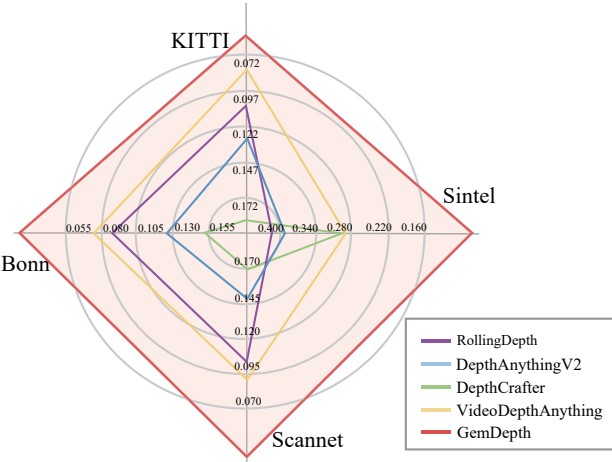

*Figure 2.* **Leaderboard performance.** The radar chart illustrates that GemDepth ranks first across all four benchmark video datasets, significantly advancing the state-of-the-art. The reported metric is the AbsRel error.

inter-frame flickering and geometric inconsistencies.

Current mainstream solutions generally fall into two categories. The first category leverages generative frameworks, such as video diffusion models (Blattmann et al., 2023a;b; Chen et al., 2023; 2024; Yang et al., 2024a; Hu et al., 2025; Shao et al., 2025; Ke et al., 2025) to infer depth sequences. While these models excel at preserving sharp structural details, they suffer from poor temporal coherence and incur heavy computational overheads, including prohibitive memory usage and slow inference speeds. The second category consists of discriminative approaches (Wang et al., 2022; Yasarla et al., 2023; 2024; Kuang et al., 2025; Chen et al., 2025; Sobko et al., 2026) that extend single-frame estimators to the video domain via auxiliary temporal modules. Despite their efficiency, these models face a fundamental limitation: they primarily rely on implicit temporal smoothing on 2D frames, which lacks the geometric awareness necessary for genuine 3D spatio-temporal consistency. This makes them struggle to maintain strict geometric coherence, particularly during complex camera rotations or drastic view changes. Furthermore, by prioritizing global smoothness, these frameworks often suppress high-frequency spatial cues, resulting in blurred boundaries and degraded structural fidelity. We argue that explicit 3D geometry understanding—encompassing the perception of camera motion and global 3D structure—is a prerequisite for achieving genuine temporal consistency. Lacking explicit motion-aware priors, models fail to establish the latent point-level correspondences necessary to enforce consistency. Consequently, they become susceptible to interference from inconsistent temporal cues, leading to spatial blurring.

To bridge this gap, we present GemDepth, a framework that enhances video depth estimation through two synergistic innovations. First, the Geometry-Embedding Module (GEM)

explicitly models camera ego-motion to inject pose-aware geometric embeddings. By processing a learnable camera token, GEM estimates 6-DoF poses and a global scale factor, which are normalized into a unified frame. These components are encoded via MLPs and fused with visual features. This process transforms abstract motion into concrete geometric embeddings, equipping the network with a metric-aware coordinate system that enforces consistency through physical constraints rather than blind smoothing. As shown in Fig. 3, this ensures robust 3D structural integrity even under large ego-motion. Second, the Alternating Spatio-Temporal Transformer (ASTT) integrates temporal pose embeddings to decouple dependency modeling into two specialized phases: (1) Temporal Attention, which leverages GEM's motion priors to establish explicit point-level correspondences, ensuring robust spatial structural consistency; and (2) Spatial Attention, which aggregates relevant 3D spatial features to enhance high-frequency representations. By alternating between geometric alignment and detail refinement, ASTT yields depth sequences that are both high-fidelity and flicker-free.

Notably, GemDepth is a generalized framework seamlessly integrable into various monocular or video depth baselines. To ensure optimal convergence, we adopt a two-stage training strategy: first, jointly optimizing all parameters via pose and depth supervision to establish geometry-consistent motion representations; second, freezing the GEM module to fine-tune the remaining components. This compels ASTT to adapt to pose noise, yielding high-quality video depth even with imperfect geometric cues. This data-efficient paradigm ensures superior performance using only limited pose-labeled data. Extensive evaluations across diverse datasets demonstrate that GemDepth achieves state-of-the-art (SOTA) zero-shot performance, setting new benchmarks for both spatial accuracy and temporal consistency. Our main contributions are summarized as follows:

- We propose GemDepth, a generalized framework that incorporates a Geometry-Embedding Module (GEM) to inject pose-aware embeddings, enforcing rigorous 3D structural constraints and consistency.

- We introduce the ASTT module, which leverages geometric priors to decouple dependency modeling. By alternating between geometrically-guided temporal alignment and spatial refinement, ASTT resolves the conflict between detail preservation and temporal stability.

- GemDepth is a generalized solution that consistently improves diverse baselines. It achieves SOTA performance with highly efficient data utilization, surpassing existing approaches while using minimal training samples.

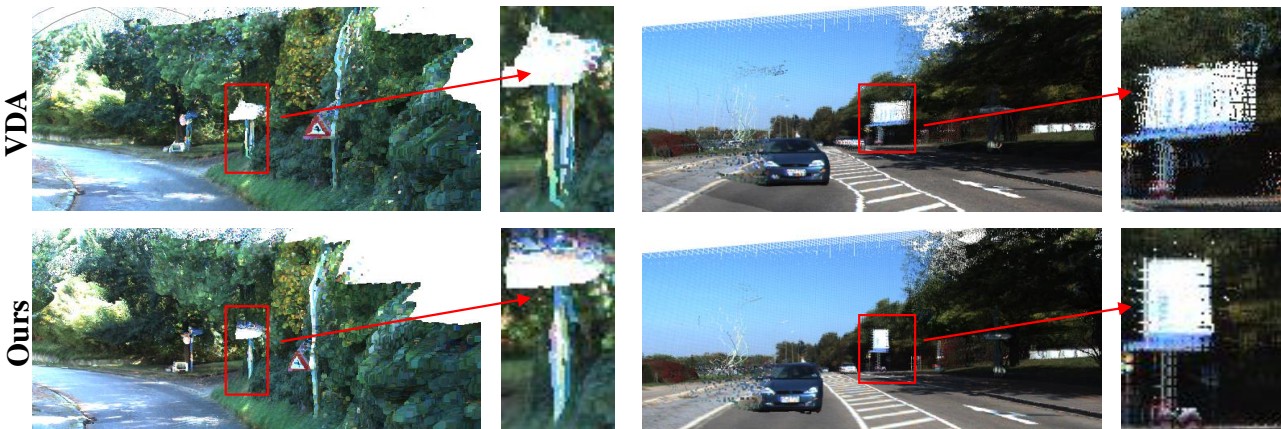

*Figure 3.* **Visualization of zero-shot point clouds on KITTI.** Accumulated from 10 consecutive frames using GT poses, GemDepth demonstrates superior 3D temporal consistency, maintaining structural integrity even under camera rotation and large ego-motion.

## 2. Related Work

### 2.1. Monocular Depth Estimation

Monocular depth estimation (MDE) aims to predict depth maps from single images (Fu et al., 2018; Bhat et al., 2021; Yuan et al., 2022; Li et al., 2023; Wang et al., 2025b; 2026). Despite the substantial progress achieved by recent deep learning approaches, achieving robust generalization across diverse scenes remains a persistent challenge. To address this, MiDaS (Birkl et al., 2023) introduced affine-invariant alignment for multi-dataset training, a paradigm further scaled by DepthAnythingV2 (Yang et al., 2024b) through massive semi-supervised learning. More recent innovations have further elevated structural fidelity and metric accuracy: Pixel-Perfect Depth (Xu et al., 2025) integrates semantic priors via Diffusion Transformers, while Depth Pro (Bochkovskii et al., 2024) leverages DINOv2 (Oquab et al., 2023) to achieve zero-shot metric depth with crisp structural details. However, as these models are primarily trained on static images, they lack intrinsic temporal constraints, leading to severe instability and flickering when applied to video sequences.

### 2.2. Video Depth Estimation

Existing video depth estimation methods are primarily categorized into discriminative and generative models. Discriminative models, such as NVDS (Wang et al., 2023), Buffer Anytime (Kuang et al., 2025), and VideoDepthAnything (Chen et al., 2025), prioritize efficiency via direct prediction or lightweight temporal attention. Recently, StableDPT (Sobko et al., 2026) utilized keyframe integration to improve sequence stability. However, these methods often suffer from blurring or flickering due to a lack of intrinsic 3D perception and alignment. Generative models like RollingDepth (Ke et al., 2025), DepthCrafter (Hu et al., 2025), and ChronoDepth (Shao et al., 2025) leverage video

diffusion for high-precision spatial predictions but are hindered by slow inference and high memory costs. In this work, we build GemDepth upon discriminative baselines, integrating 3D geometric priors and spatio-temporal modeling to yield superior video depth. By establishing latent point-level correspondences, GemDepth achieves sharper spatial details and robust temporal consistency with high efficiency.

## 3. Method

GemDepth is a generalized framework capable of enhancing diverse architectures. We instantiate it on two baselines: DepthAnythingV2 (Yang et al., 2024b) and VideoDepthAnything (Chen et al., 2025). As the latter primarily extends the DPT decoder with a temporal head, we adopt DepthAnythingV2 as the representative example for our method description. Our method is organized as follows: Sec. 3.1 outlines the feature extraction process. Sec. 3.2 introduces the Geometry-Embedding Module (GEM), which constructs 3D geometric priors to guide depth prediction. Sec. 3.3 details the Alternating Spatio-Temporal Transformer (ASTT). Finally, Sec. 3.4 and Sec. 3.5 specify the loss function and the two-stage training paradigm, respectively.

### 3.1. Feature Extraction

Given a video input $\mathbf{X} \in \mathbb{R}^{B \times N \times C \times H \times W}$, we collapse the temporal dimension $N$ into the batch dimension $B$ to form $\hat{\mathbf{X}} \in \mathbb{R}^{(B \times N) \times C \times H \times W}$, $C, H, W$ correspond to the channel, height, and width dimensions. A frozen DINOv2 (Oquab et al., 2023) processes $\hat{\mathbf{X}}$ by flattening non-overlapping $p \times p$ patches, extracting multi-scale features $F_j \in \mathbb{R}^{(B \times N) \times \frac{HW}{p^2} \times C_j}$ from layers $j \in \{5, 12, 18, 24\}$. Since this frame-wise encoding lacks temporal interaction, we subsequently introduce the GEM and ASTT modules to

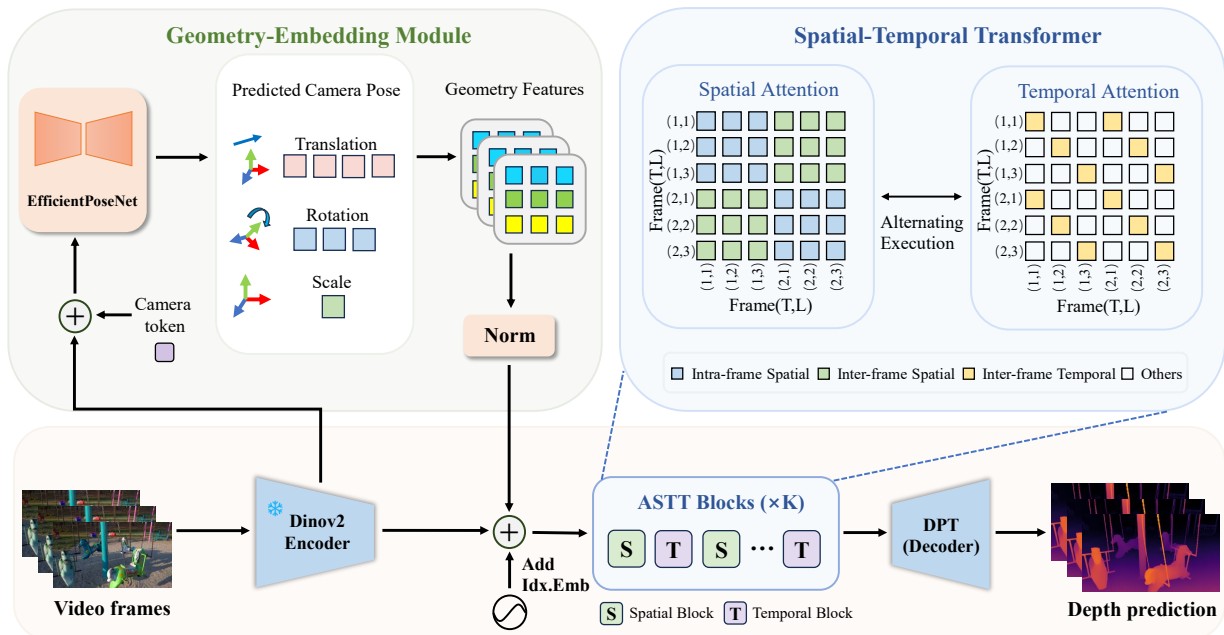

*Figure 4.* **Overview of GemDepth-DAv2.** Built upon the ViT-based encoder and DPT head of DepthAnythingV2 (Yang et al., 2024b), GemDepth-DAv2 incorporates two novel components: Geometry-Embedding Module (GEM) and Alternating Spatio-Temporal Transformer (ASTT). By synergistically aggregating 3D geometric constraints and multi-scale spatio-temporal interactions, GemDepth-DAv2 effectively addresses the long-standing inconsistency issues in video depth estimation caused by the lack of geometric alignment.

inject geometric and spatio-temporal consistency.

## 3.2. Geometry-Embedding Module (GEM)

The Geometry-Embedding Module (GEM) recovers 3D geometric consistency by transforming abstract camera ego-motion into concrete structural embeddings. Built upon a lightweight EfficientPoseNet (Wang et al., 2025a), it consists of a 4-layer alternating-attention transformer that processes a learnable camera token $\mathbf{t} \in \mathbb{R}^{(B \times N) \times 1 \times D}$ injected into the $F_4$ feature map. This transformer fuses intra-frame spatial context with inter-frame temporal motion cues to obtain the refined token $\hat{\mathbf{t}}$, which is then passed through a pose head to estimate 6-DoF camera poses $\mathbf{g} \in \mathbb{R}^{(B \times N) \times 7}$. To enforce physical constraints, all poses are projected into a unified canonical frame (Wang et al., 2024). To resolve monocular scale ambiguity, we normalize per-frame translations as $\hat{T}_i = \frac{T_i}{Z}$ using a global scale factor $Z = \frac{1}{N} \sum_{i=1}^{N} \|T_i\|$. By explicitly supervising the GEM module with scale-normalized ground truth, we ensure the predicted poses inherently operate within a unified scale space, thereby eliminating any initial scale inconsistencies. The resulting components—rotation $\mathbf{Q} \in \mathbb{R}^{(B \times N) \times 4}$, translation $\hat{\mathbf{T}} \in \mathbb{R}^{(B \times N) \times 3}$, and scale factor $\mathbf{Z} \in \mathbb{R}^{(B \times N) \times 1}$—are encoded via geometric MLPs (Keetha et al., 2025) to generate the camera feature representation $F_{cam} \in \mathbb{R}^{(B \times N) \times D}$. This metric-aware embedding is finally fused with the primary feature maps $F_4$, providing explicit geometric guidance that steers depth refinement through physical constraints rather

than blind smoothing.

## 3.3. Alternating Spatio-Temporal Transformer (ASTT)

The Alternating Spatio-Temporal Transformer (ASTT) unifies spatio-temporal representations by decoupling dependency modeling into two strategic phases. To encode precise positional relationships, the enhanced features from GEM are integrated with RoPE (Su et al., 2024) for spatial encoding and an inter-frame Idx Embedding (Yang et al., 2025) for chronological order. Given the input enhanced visual features $\hat{F} \in \mathbb{R}^{(B \times N) \times L \times D}$, ASTT alternates between temporal alignment and spatial refinement to enforce geometric consistency. (1) Temporal Attention for Geometric Alignment: We first reorganize the feature maps to explicitly isolate temporal dependencies at aligned spatial positions, as illustrated in Fig. 4. Leveraging GEM's 6-DoF motion priors, this phase establishes explicit point-level correspondences across the temporal axis. By performing trajectory-based feature aggregation, it captures pure motion cues while mitigating interference from complex spatial contexts, ensuring robust spatial structural consistency and flicker-free depth sequences. (2) Spatial Attention for Structural Refinement: conditioned on the aligned temporal features, we restructure the representations to perform global information exchange. As shown in Fig. 4, this process is decomposed into two specialized mechanisms: intra-frame spatial attention captures local layout and inter-frame spatial attention models long-range dependencies across frames. This phase aggregates

relevant 3D spatial features to enhance high-frequency representations and sharpen structural details. By alternating between these two mechanisms, ASTT enforces geometric consistency before detail sharpening, delivering high-fidelity depth sequences. In Sec. 4.5, we demonstrate that both spatial and temporal attention modules play a significant role when leveraging geometric priors.

### 3.4. Loss Function

To effectively supervise the GEM module, we supervise the GEM module via a Huber-based camera loss applied to rotation and translation following (Wang et al., 2025a), formulated as:

$$\mathcal{L}_{\text{rot}} = ||\hat{Q}_i - Q_i||_\epsilon \qquad \mathcal{L}_{\text{trans}} = ||\hat{T}_i - T_i||_\epsilon \qquad (1)$$

Specifically, $Q_i$ and $\hat{Q}_i$ denote the predicted and ground-truth rotations, while $T_i$ and $\hat{T}_i$ represent the predicted and ground-truth normalized translations, respectively. The rotation loss $\mathcal{L}_{\text{rot}}$ and translation loss $\mathcal{L}_{\text{trans}}$ are computed using the Huber loss $|| \cdot ||_\epsilon$.

$$\mathcal{L}_{\text{cam}} = \frac{1}{N} \sum_{i=1}^{N} (\mathcal{L}_{\text{rot}} + \lambda \mathcal{L}_{\text{trans}}) \qquad (2)$$

Finally, $\lambda$ is employed as a balancing weight between these two loss terms.

For depth supervision, we employ the scale- and shift-invariant loss $\mathcal{L}_{\text{ssi}}$ and the multi-scale gradient matching loss $\mathcal{L}_{\text{gm}}$ following MiDaS (Birkl et al., 2023) . Furthermore, we introduce the $\mathcal{L}_{\text{tgm}}$ loss from VideoDepthAnything (Chen et al., 2025) to enhance temporal geometric consistency.

The total loss for video depth is formulated as:

$$\mathcal{L}_{\text{total}} = \mathcal{L}_{\text{ssi}} + \alpha \mathcal{L}_{\text{gm}} + \beta \mathcal{L}_{\text{tgm}} + \gamma \mathcal{L}_{\text{cam}} \qquad (3)$$

where the weighting coefficients are set to $\alpha = 0.5$, $\beta = 10$, and $\gamma = 0.2$.

### 3.5. Two Stage Training

Training models with geometric constraints inherently relies on large-scale video sequences paired with ground-truth camera poses. However, such pose annotations are scarce in most existing datasets. To mitigate this data bottleneck, we employ a two-stage training strategy that decouples the optimization of geometric components from general depth estimation.

Stage 1: **Geometric Optimization**. In this initial phase, we jointly optimize the entire framework by enforcing supervision on both the pose estimates from GEM and the final depth predictions. This dual supervision strategy effectively aligns motion representations with geometric constraints.

We utilize a diverse composite dataset comprising Virtual KITTI 2 (Cabon et al., 2020), TartanAir (Wang et al., 2020), PointOdyssey (Zheng et al., 2023), MVS-Synth (Huang et al., 2018), and Dynamic Replica (Karaev et al., 2023), totaling approximately 690K frames. This stage effectively instills robust geometric reasoning capabilities, enabling the model to achieve state-of-the-art performance in video depth estimation.

Stage 2: **Depth Refinement**. Once the GEM module has converged, we freeze its weights to retain the established geometric foundation. The rest of the network is then fine-tuned using roughly 250K frames from datasets without camera extrinsics ground-truth, such as IRS (Wang et al., 2019) and diverse in-the-wild sequences. Comprehensive details for all training datasets are provided in A.2. This strategy effectively enhances the robustness of the ASTT against pose uncertainties, enabling consistent high-quality depth prediction even in the presence of imperfect geometric guidance.

## 4. Experiment

### 4.1. Implementation Details

GemDepth is implemented based on DepthAnythingV2 (DAv2) and VideoDepthAnything (VDA), resulting in two variants: GemDepth-DAv2 and GemDepth-VDA. We employ the large-scale variants (ViT-Large) of two methods, thereby benchmarking against their maximum capacity. We train the models on video clips of length $N = 32$ using a multi-resolution strategy (Keetha et al., 2025) with a base resolution of $518 \times 518$. The models are optimized via AdamW with a discriminative learning rate: $1 \times 10^{-4}$ for the newly introduced ASTT and GEM modules, and $1 \times 10^{-6}$ for the pre-trained components. Training is conducted on 16 NVIDIA A800 GPUs with a global batch size of 16. Each phase of our two-stage training paradigm reaches convergence in approximately three days.

### 4.2. Evaluation

**Evaluation Datasets.** To quantitatively evaluate the performance and generalization of GemDepth, we conduct extensive experiments on four representative benchmarks encompassing a diverse range of indoor and outdoor environments: Sintel (Butler et al., 2012), KITTI (Geiger et al., 2013), Scannet (Dai et al., 2017), and Bonn (Palazzolo et al., 2019). Following the evaluation protocol established in VideoDepthAnything (Chen et al., 2025), we limit the test sequences to a maximum of 500 frames for long video depth estimation.

**Evaluation Metrics.** We perform affine-invariant alignment (Dong et al., 2022; Yang et al., 2024b) by computing the optimal scale and shift between predictions and ground

*Table 1.* **Zero-shot depth estimation results.** We benchmark our method against state-of-the-art video depth estimation approaches. GemDepth-DAv2 and GemDepth-VDA denote our model variants instantiated on DepthAnythingV2 (Yang et al., 2024b) and VideoDepthAnything (Chen et al., 2025), respectively. The best and second best are marked with colors.

| Method | Sintel (~50 frames) | | Bonn (500 frames) | | Scannet (500 frames) | | KITTI (500 frames) | | Runtime |
|---|---|---|---|---|---|---|---|---|---|
| | Absrel↓ | $\delta_1$↑ | Absrel↓ | $\delta_1$↑ | Absrel↓ | $\delta_1$↑ | Absrel↓ | $\delta_1$↑ | (ms) |
| NVDS | 0.408 | 0.464 | 0.199 | 0.674 | 0.207 | 0.628 | 0.233 | 0.614 | 258 |
| ChronoDepth | 0.192 | 0.673 | 0.199 | 0.665 | 0.199 | 0.665 | 0.243 | 0.576 | 617 |
| DepthCrafter | 0.299 | 0.695 | 0.153 | 0.803 | 0.169 | 0.730 | 0.164 | 0.753 | 980 |
| RollingDepth | 0.417 | 0.375 | 0.088 | 0.931 | 0.102 | 0.901 | 0.107 | 0.887 | 280 |
| DepthAnythingV2 | 0.390 | 0.541 | 0.127 | 0.864 | 0.150 | 0.768 | 0.137 | 0.815 | **79** |
| VideoDepthAnything | 0.295 | 0.644 | 0.071 | 0.959 | 0.089 | 0.926 | 0.083 | 0.944 | 85 |
| GemDepth-DAv2(Ours) | 0.188 | 0.812 | 0.055 | 0.970 | 0.069 | 0.959 | 0.077 | 0.950 | 94 |
| GemDepth-VDA(Ours) | **0.157** | **0.827** | **0.051** | **0.978** | **0.066** | **0.967** | **0.071** | **0.955** | 99 |

*Table 2.* **Quantitative evaluation of temporal consistency on Scannet (Dai et al., 2017).** We assess the inter-frame stability of depth sequences using the Temporal Alignment Error (TAE) metric. The **best** result is bolded, and the second-best is underlined.

| Method | DepthAnythingV2 | RollingDepth | VideoDepthAnything | GemDepth-DAv2 | GemDepth-VDA |
|---|---|---|---|---|---|
| TAE ↓ | 1.14 | 0.65 | 0.57 | 0.50 (−56.14%) | **0.47** (−17.54%) |

truth. To evaluate the performance, we employ six key metrics spanning four distinct aspects: Absolute Relative Error (AbsRel) and the $\delta_1$ accuracy (Yang et al., 2024b) are used to assess spatial precision; Temporal Alignment Error (TAE) (Yang et al., 2024a) and Temporal Chamfer Distance (TCD) are introduced to measure temporal consistency across frames; the F1 (Lin et al., 2025) score is utilized to evaluate point cloud reconstruction quality; and Absolute Trajectory Error (ATE) is adopted to measure camera pose accuracy. Detailed definitions and calculation procedures for these metrics are provided in Sec. A.3.

### 4.3. Zero-shot Depth Estimation

We evaluate GemDepth against several representative video depth estimation frameworks, including NVDS (Wang et al., 2023), ChronoDepth (Shao et al., 2025), DepthCrafter (Hu et al., 2025), and RollingDepth (Ke et al., 2025). To explicitly quantify the performance gains of our framework, we also report results for DepthAnythingV2 and VideoDepthAnything as comparative baselines.

**Spatial Precision.** As demonstrated in Tab. 1, GemDepth consistently establishes a new state-of-the-art across all metrics, irrespective of whether it is instantiated on DepthAnythingV2 or VideoDepthAnything. Notably, GemDepth-VDA achieves an AbsRel error reduction of over 25% on the Bonn and Scannet datasets compared to existing SOTA methods. This performance gap is even more pronounced on the Sintel dataset, where our approach surpasses VideoDepthAnything by a substantial 46.8%. This wide margin is particularly revealing, as it highlights a critical disparity: while rapid camera motion and drastic viewpoint shifts often confound leading baselines,

GemDepth leverages explicit geometric priors to accurately resolve these challenging spatial transformations. Even our GemDepth-DAv2 outperforms the backbone-equivalent VideoDepthAnything. Since both models utilize the same pre-trained encoder, this performance gap effectively isolates the contribution of our proposed GEM and ASTT modules, demonstrating their distinct advantage in improving spatial precision and temporal coherence. Crucially, GemDepth achieves these results with superior data efficiency: training on fewer than 1M frames, compared to the 1.3M and >10M frames required by VideoDepthAnything and DepthCrafter, respectively. Additional evaluations on varying sequence lengths are provided in B.2.

**Temporal Consistency.** Tab. 2 presents a quantitative evaluation of temporal consistency. Benchmarked on the first 20 sequences of the Scannet dataset (110 frames per sequence), GemDepth consistently yields the most stable depth estimations. Notably, both GemDepth-DAv2 and GemDepth-VDA establish new state-of-the-art standards for temporal stability, surpassing their respective baselines by 56.14% and 17.54% in the TAE metric. By leveraging explicit 3D geometric constraints, our model effectively suppresses the temporal flickering and erratic fluctuations prevalent.

**Qualitative Results.** Fig. 6 presents qualitative comparisons across diverse indoor and outdoor benchmarks (Sintel, Bonn, and Scannet). As highlighted by the white arrows, GemDepth-VDA demonstrates superior spatial precision and structural fidelity, effectively recovering fine-grained details while mitigating the over-smoothing artifacts typical of competing methods. Notably, the second row illustrates our model's capability to reconstruct challenging dynamic objects (e.g., airborne balloons) with sharp, distinct con-

*Table 3.* **Quantitative comparison on three datasets.** AbsRel/TAE evaluate depth quality; ATE (trajectory error) and F1 (F-Score) measure 3D structural fidelity.

| Method | Params | Scannet | | | | Sintel | | | | Bonn | | | |
|---|---|---|---|---|---|---|---|---|---|---|---|---|---|
| | | ATE ↓ | F1 ↑ | Absrel↓ | TAE ↓ | ATE ↓ | F1 ↑ | Absrel↓ | TAE ↓ | ATE ↓ | F1 ↑ | Absrel↓ | TAE ↓ |
| VGGT | 1.10B | **0.02** | 65.46 | 0.13 | 1.92 | **0.02** | 70.47 | 0.55 | 1.18 | **0.02** | 76.58 | 0.20 | 2.38 |
| DA3 | 1.19B | **0.02** | 65.91 | 0.11 | 1.12 | **0.02** | 71.91 | 0.38 | 1.37 | **0.02** | 78.44 | 0.18 | 2.87 |
| GemDepth | 0.58B | 0.03 | **69.01** | **0.07** | **0.47** | 0.03 | **72.66** | **0.16** | **0.84** | 0.03 | **90.43** | **0.05** | **2.03** |

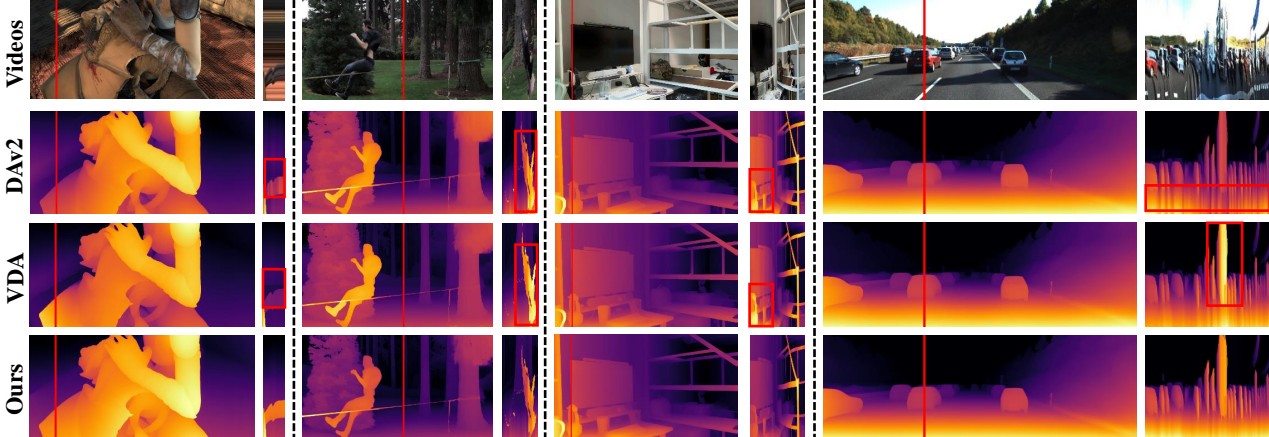

*Figure 5.* **Qualitative results of temporal consistency on videos of varying lengths.** We compared GemDepth-VDA with DepthAnythingV2 (Yang et al., 2024b) and VideoDepthAnything (Chen et al., 2025) using sequences of increasing lengths from Sintel (Butler et al., 2012), DAVIS (Perazzi et al., 2016), KITTI (Geiger et al., 2013), and in-the-wild datasets. The red boxes highlight the temporally inconsistent depth estimations.

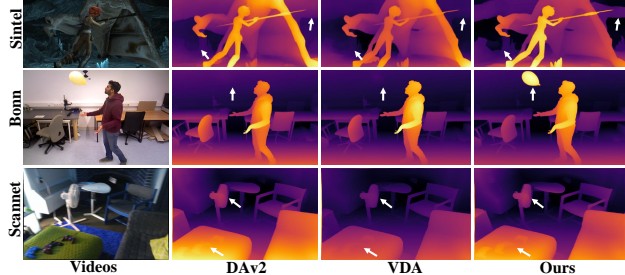

*Figure 6.* **Qualitative comparison of spatial accuracy on Sintel (Butler et al., 2012), Bonn (Palazzolo et al., 2019), and Scannet (Dai et al., 2017).** As indicated by the white arrows, GemDepth-VDA outperforms existing approaches in recovering background depth and preserving fine structural details.

tours. This performance is directly attributed to our robust structural consistency, which explicitly preserves the geometric integrity of dynamic subjects.

To assess temporal stability, we visualize temporal profiles in Fig. 5 by extracting depth slices along a fixed spatial axis (indicated by the red line). GemDepth-VDA demonstrates exceptional temporal coherence, maintaining seamless transitions across both short and long sequences. In contrast, DepthAnythingV2 and VideoDepthAnything suffer from pronounced flickering and jagged temporal discontinuities.

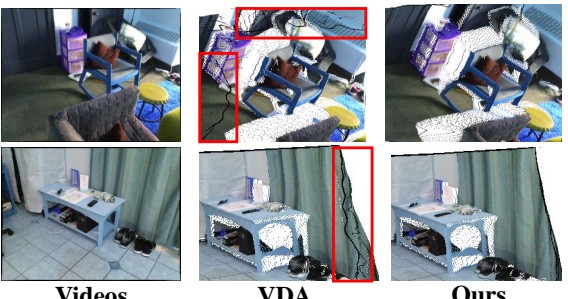

*Figure 7.* **Qualitative comparison of point clouds on Scannet (Dai et al., 2017).** We compare GemDepth-VDA with VideoDepthAnything. The red boxes highlight areas of disorganized streaking at object boundaries.

Furthermore, we assess 3D geometric fidelity via point cloud reconstruction in Fig. 7. GemDepth-VDA generates clean, well-structured surfaces, whereas competing methods suffer from disorganized streaking at object boundaries.

**Inference cost.** To ensure a fair comparison, we evaluate the inference latency of all models on a single NVIDIA A800 GPU at a resolution of $518 \times 518$. As summarized in Tab. 1, GemDepth achieves substantially lower latency than generative frameworks. When compared to discriminative baselines (DepthAnythingV2 and VideoDepthAnything), GemDepth delivers superior spatio-temporal precision while

*Table 4.* **Ablation study of the proposed modules on KITTI, Sintel, and Scannet.** All variants are built upon VideoDepthAnything(VDA) (Chen et al., 2025) and trained for 20k steps during Stage 1. The **best** result is bolded.

| Model | Components | | | KITTI | Sintel | Scannet | Run-time |
|---|---|---|---|---|---|---|---|
| | GEM | Spatial | Temporal | Absrel↓ | Absrel↓ | TAE↓ | (ms) |
| Baseline (VDA) | × | × | × | 0.092 | 0.356 | 0.621 | 85 |
| Baseline + Spatial | × | ✓ | × | 0.082 | 0.337 | 0.609 | 90 |
| Baseline + Temporal | × | × | ✓ | 0.084 | 0.343 | 0.573 | 88 |
| Baseline + ASTT | × | ✓ | ✓ | 0.080 | 0.328 | 0.566 | 95 |
| Full Model | ✓ | ✓ | ✓ | **0.074** | **0.295** | **0.538** | 99 |

*Table 5.* **Robustness evaluation under varying levels of injected pose noise on the Scannet dataset.**

| Add Noise | 0% | 5% | 10% | 20% | 50% | 100% |
|---|---|---|---|---|---|---|
| AbsRel | 0.066 | 0.066 | 0.067 | 0.071 | 0.078 | 0.096 |
| TAE | 0.472 | 0.475 | 0.480 | 0.503 | 0.577 | 0.638 |

incurring a marginal computational overhead of less than 20% for both variants.

### 4.4. 3D Geometric Accuracy

**Evaluation on diverse Scenes.** We evaluate GemDepth-VDA against 3D foundation models DA3 (Lin et al., 2025) and VGGT (Wang et al., 2025a). For a fair comparison, we follow VDA protocols and uniformly provide 32 consecutive views to all models. As shown in Tab. 3, GemDepth-VDA significantly outperforms DA3 in video depth estimation, cutting the TAE on Scannet by more than half (0.47 vs. 1.12) and reducing AbsRel by 70% on Bonn (0.05 vs. 0.18). Moreover, our intermediate module predicts camera poses (ATE) comparable to these heavy foundation models. This reliable pose, combined with our accurate depth, results in superior 3D structural fidelity (F1), establishing new state-of-the-art scores on Bonn (90.43 vs. 78.44). Remarkably, GemDepth-VDA achieves this comprehensive 3D consistency with only 0.58B parameters—roughly half that of DA3 or VGGT—proving the high efficiency of our geometry-guided design.

### 4.5. Ablation Studies

**Effectiveness of the ASTT module.** To validate the effectiveness of our proposed modules, we conduct ablation experiments across the KITTI, Sintel, and Scannet datasets. As shown in Tab. 4, the independent spatial and temporal components play distinct yet complementary roles. Specifically, the 'Baseline + Spatial' configuration primarily drives improvements in spatial precision, noticeably reducing the AbsRel error on KITTI (0.092 → 0.082) and Sintel (0.356 → 0.337). Conversely, the 'Baseline + Temporal' variant is

highly effective at enhancing inter-frame stability, delivering a much sharper reduction in temporal inconsistency on Scannet (TAE drops from 0.621 to 0.573) compared to its spatial counterpart. When integrated into the unified ASTT framework ('Baseline + ASTT'), these modules exhibit a powerful synergistic effect, simultaneously optimizing both depth accuracy and temporal coherence. Furthermore, this comprehensive capability is achieved highly efficiently, with the ASTT integration introducing a modest 10ms increase in inference latency (85ms → 95ms).

Moreover, to comprehensively evaluate the robustness of the ASTT module against potential pose inaccuracies, we conducted additional experiments. Tab. 5 evaluates the internal robustness of our framework by injecting relative Gaussian noise—scaled to the original translation and quaternion magnitudes—directly into the intermediate camera poses self-predicted by our GEM module. Evaluated on the Scannet dataset, GemDepth maintains highly stable performance under low-to-moderate noise conditions. Notably, as the noise level scales from 0% up to 20%, the performance drop is minimal, with AbsRel merely shifting from 0.066 to 0.071 and TAE from 0.472 to 0.503. Furthermore, the model exhibits a graceful degradation, showing significant impact only when the injected noise exceeds 50%. This confirms that our downstream depth prediction is highly resilient to upstream geometric inaccuracies. It demonstrates that GemDepth effectively leverages self-predicted poses for 3D alignment without becoming catastrophically over-reliant on them, ensuring robust performance even if internal pose estimations fluctuate in complex scenarios.

**Effectiveness of the GEM module.** We further investigate the impact of the GEM module by comparing 'Baseline + ASTT + GEM' (Full Model) against 'Baseline + ASTT'. As reported in Tab. 4, 'Baseline + ASTT + GEM' yields an additional 7.5% and 10.1% improvement in the AbsRel on KITTI and Sintel, respectively. Furthermore, this module enhances temporal stability on Scannet, providing an 4.9% gain in the TAE metric. This integration introduces a negligible increase of 5ms in inference latency, bringing the total runtime to 99ms.

*Table 6.* **Effectiveness of our two-stage training strategy on four datasets.** We evaluate the performance of our model at the end of different training stage and compare it with VideoDepthAnything (Chen et al., 2025). The **best** result is bolded.

| Method | Stage | Data | KITTI | | Sintel | | Bonn | | Scannet | | |
|---|---|---|---|---|---|---|---|---|---|---|---|
| | | | AbsRel ↓ | $\delta_1$ ↑ | AbsRel ↓ | $\delta_1$ ↑ | AbsRel ↓ | $\delta_1$ ↑ | AbsRel ↓ | $\delta_1$ ↑ | TAE ↓ |
| VDA | - | 1.3M | 0.083 | 0.944 | 0.295 | 0.644 | 0.071 | 0.959 | 0.089 | 0.926 | 0.57 |
| GemDepth-DAv2 | Stage 1 | 690k | 0.086 | 0.938 | 0.295 | 0.625 | 0.080 | 0.930 | 0.080 | 0.950 | 0.57 |
| | Stage 2 | 940k | 0.077 | 0.950 | 0.188 | 0.811 | 0.055 | 0.970 | 0.069 | 0.959 | 0.50 |
| GemDepth-VDA | Stage 1 | 690k | 0.077 | 0.950 | 0.278 | 0.638 | 0.065 | 0.965 | 0.078 | 0.949 | 0.59 |
| | Stage 2 | 940k | **0.071** | **0.955** | **0.157** | **0.827** | **0.055** | **0.970** | **0.066** | **0.967** | **0.47** |

*Table 7.* **Ablation study of the place for ASTT module on KITTI, Sintel, and Scannet.** The **best** result is bolded.

| Method | KITTI | | Sintel | | Scannet |
|---|---|---|---|---|---|
| | Absrel ↓ | $\delta1$ ↑ | Absrel ↓ | $\delta1$ ↑ | TAE ↓ |
| Late-stage (Position 3) | 0.126 | 0.842 | 0.377 | 0.557 | 0.917 |
| Mid-stage (Position 2) | 0.102 | 0.913 | 0.352 | 0.586 | 0.737 |
| Early-stage (Position 1) | **0.088** | **0.938** | **0.328** | **0.599** | **0.654** |

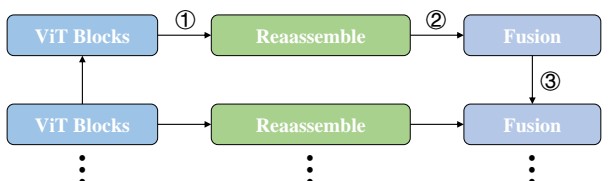

*Figure 8.* **Placement of ASTT across different feature processing stages.**

**Effectiveness of the two-stage training strategy.** To validate the efficacy of our proposed two-stage training strategy, we assess model performance upon the completion of each phase on four benchmarks. As detailed in Tab. 6, we observe a consistent upward trajectory for both GemDepth-DAv2 and GemDepth-VDA: performance monotonically improves as training advances from Stage 1 to Stage 2.

**Place for ASTT module.** Crucially, the effectiveness of the ASTT module is highly sensitive to its architectural placement. As illustrated in Fig. 8, we compare three strategies: Position 1 (Early-stage) adopted by GemDepth, Position 2 (Mid-stage) used by VideoDepthAnything (Chen et al., 2025), and Position 3 (Late-stage) utilized by FlashDepth (Chou et al., 2025). Unlike existing methods that perform temporal modeling on abstracted features within the DPT decoder, our ASTT module executes spatio-temporal interaction immediately following feature extraction. As reported in Tab. 7, experimental results demonstrate the clear superiority of our early-stage interaction strategy. Compared to Position 3 and Position 2, Position 1 reduces AbsRel by 30.2% and 13.7% on KITTI, and by 13.0% and

6.8% on Sintel. Furthermore, the TAE on Scannet drops by 28.7% and 11.3% respectively. These gains confirm that early-stage interaction is vital for capturing fine-grained geometric details before features become overly abstracted in deeper decoder layers.

## 5. Conclusion

In this paper, we propose GemDepth, a novel and versatile framework for video depth estimation. Existing approaches often suffer from severe temporal flickering and scale ambiguity when applied to highly dynamic scenes or long-term videos. The core components of our method include: (i) Geometry-Embedding Module (GEM), which explicitly injects 3D geometric constraints into features to ensure structural consistency and suppress flickering; (ii) Alternating Spatio-Temporal Transformer (ASTT), which synergizes spatio-temporal cues to refine inter-frame correspondences and boundary clarity; and (iii) a decoupled two-stage training strategy that enables effective geometric optimization on datasets lacking camera parameters. GemDepth establishes a new state-of-the-art across four major benchmarks and demonstrates robust zero-shot generalization to diverse real-world sequences of varying lengths.

## Impact Statement

This paper presents a framework that enhances the spatial accuracy and temporal consistency of video depth estimation. By striking a favorable balance between high-precision perception and computational feasibility, our work improves the practicality of depth estimation backbones for a wide array of real-world applications, such as autonomous driving, robotic navigation, and augmented reality. Any societal consequences or impacts typically associated with the advancement of robust machine perception apply here, as GemDepth facilitates the reliable deployment of 3D sensing in complex environments and supports the high-quality processing of video streams for safety-critical visual tasks.

## Acknowledgement

This research is supported by National Natural Science Foundation of China (U25B2045, 62472184) and the Fundamental Research Funds for the Central Universities.

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

# A. Implementation Details

## A.1. Data Preparation

Following DepthAnythingV2 (Yang et al., 2024b), we first transform the depth values $d$ into the disparity space via $s = 1/d$, followed by a normalization to $[0, 1]$. We then employ the affine-invariant loss (Birkl et al., 2023) to account for the unknown scale and shift inherent in each sample. For datasets provided with camera intrinsics, we apply corresponding resizing and translation operations to align them with the training resolution.

## A.2. Training Dataset Details

As illustrated in the Tab. 8, our training set comprises a total of 940K image pairs, consisting of 690K samples from pose-annotated datasets and 250K from pose-free datasets. Notably, as only a subset of the PointOdyssey dataset contains ground-truth background depth, we follow FlashDepth (Chou et al., 2025) by utilizing only this specific portion for training. To ensure a balanced representation during the optimization process, we assign appropriate sampling weights to each dataset, achieving a uniform sampling probability across all sources.

Due to the inconsistent depth ranges across different datasets, our experiments indicate that the absence of a sky mask significantly degrades the clarity of sky predictions. To address this, we truncate the depth values for sky regions in all outdoor datasets to 400 for supervision. For our multi-resolution training strategy, we resize the shorter edge of the images to 518 and apply random center cropping. This yields training clips with diverse resolutions, specifically $518 \times 518$, $518 \times 392$, $518 \times 336$, $518 \times 294$, $518 \times 252$, and $518 \times 168$, all with a temporal length of 32. This approach enables the network to generalize across varying aspect ratios and resolutions.

*Table 8.* **Summary of datasets used for training.**

| Dataset | Indoor | Outdoor | # Images |
|---|---|---|---|
| Pose-annotated datasets | | | |
| Virtual KITTI 2 (Cabon et al., 2020) | | ✓ | 40K |
| Tartanair (Wang et al., 2020) | ✓ | ✓ | 300K |
| PointOdyssey (Zheng et al., 2023) | ✓ | ✓ | 70K |
| MVS-Synth (Huang et al., 2018) | | ✓ | 80K |
| Dynamic Replica (Karaev et al., 2023) | ✓ | | 200K |
| Pose-free datasets | | | |
| IRS (Wang et al., 2019) | | ✓ | 100K |
| In-the-wild stereo videos | ✓ | ✓ | 150K |

## A.3. Evaluation Metric

For spatial accuracy, we align the estimated depth maps with the ground truth using a scale and shift, and calculate two metrics: AbsRel ↓ (absolute relative error: $\frac{1}{N} \sum \frac{|d - \hat{d}|}{d}$) and $\delta_1$ ↑ (percentage of $\max(\frac{d}{\hat{d}}, \frac{\hat{d}}{d}) < 1.25$).

For the temporal consistency metric(TAE), we utilize the camera intrinsics and extrinsics to perform forward and backward depth projections for calculating TAE (Yang et al., 2024a):

$$\text{TAE} = \frac{1}{2(T-2)} \sum_{k=0}^{T-1} \text{AbsRel}\left(f(\hat{x}_d^k, p^k), \hat{x}_d^{k+1}\right) + \text{AbsRel}\left(f(\hat{x}_d^{k+1}, p_-^{k+1}), \hat{x}_d^k\right), \quad (4)$$

Here, $T$ denotes the total number of frames in a sequence. The function $f$ represents the warping operation, which utilizes the transformation matrix $p^k$ (comprising both camera intrinsics and extrinsics) to project the estimated depth $\hat{x}_d^k$ from frame $k$ into the coordinate system of frame $k + 1$. Conversely, $p_-^{k+1}$ signifies the inverse transformation matrix used for backward projection. These geometric parameters are directly retrieved from the dataset to ensure accurate cross-frame alignment.

To quantitatively evaluate the 3D temporal consistency, we introduce the Temporal Chamfer Distance (TCD). Given a video sequence, we first back-project the depth maps of two consecutive frames, $t$ and $t + 1$, into 3D point clouds. Using the ground-truth camera poses, we then transform these local point clouds into the global coordinate system of the first frame, denoted as $\mathcal{P}_t$ and $\mathcal{P}_{t+1}$. The TCD is defined as the average Chamfer Distance between the aligned point clouds across the

*Table 9.* **Sensitivity analysis of GemDepth-VDA to pose integration probability.** The parameter Pose Prob. denotes the probability of feeding poses predicted by GEM into the ASTT module.

| Method | Pose Prob. | KITTI | | Sintel | | Bonn | | Scannet | |
|---|---|---|---|---|---|---|---|---|---|
| | | AbsRel $\downarrow$ | $\delta_1 \uparrow$ | AbsRel $\downarrow$ | $\delta_1 \uparrow$ | AbsRel $\downarrow$ | $\delta_1 \uparrow$ | AbsRel $\downarrow$ | $\delta_1 \uparrow$ |
| GemDepth-VDA (Baseline) | 0% | 0.088 | 0.935 | 0.278 | 0.638 | 0.065 | 0.965 | 0.078 | 0.949 |
| GemDepth-VDA (Partial) | 50% | 0.071 | 0.955 | 0.157 | 0.827 | 0.051 | 0.978 | 0.066 | 0.967 |
| GemDepth-VDA (Full) | 100% | 0.070 | 0.958 | 0.155 | 0.828 | 0.051 | 0.978 | 0.065 | 0.967 |

*Table 10.* **Quantitative comparison on the Bonn (Palazzolo et al., 2019) dataset across various video lengths (100 to 500 frames).** The **best** result is bolded.

| Method | Bonn (500frames) | | Bonn (400frames) | | Bonn (300frames) | | Bonn (200frames) | | Bonn (100frames) | |
|---|---|---|---|---|---|---|---|---|---|---|
| | AbsRel $\downarrow$ | $\delta_1 \uparrow$ | AbsRel $\downarrow$ | $\delta_1 \uparrow$ | AbsRel $\downarrow$ | $\delta_1 \uparrow$ | AbsRel $\downarrow$ | $\delta_1 \uparrow$ | AbsRel $\downarrow$ | $\delta_1 \uparrow$ |
| VideoDepthAnything | 0.071 | 0.959 | 0.069 | 0.960 | 0.067 | 0.961 | 0.063 | 0.964 | 0.045 | 0.989 |
| GemDepth-DAv2 | 0.055 | 0.970 | 0.054 | 0.970 | 0.050 | 0.971 | 0.045 | 0.973 | 0.032 | 0.993 |
| GemDepth-VDA | **0.051** | **0.978** | **0.050** | **0.977** | **0.048** | **0.979** | **0.041** | **0.982** | **0.031** | **0.995** |

sequence:

$$\text{TCD} = \frac{1}{N-1} \sum_{t=1}^{N-1} d_{\text{CD}}(\mathcal{P}_t, \mathcal{P}_{t+1}), \qquad (5)$$

where $N$ is the total number of frames in the sequence, and $d_{\text{CD}}$ represents the standard Chamfer Distance formulated as:

$$d_{\text{CD}}(\mathcal{P}_t, \mathcal{P}_{t+1}) = \frac{1}{|\mathcal{P}_t|} \sum_{\mathbf{x} \in \mathcal{P}_t} \min_{\mathbf{y} \in \mathcal{P}_{t+1}} \|\mathbf{x} - \mathbf{y}\|_2^2 + \frac{1}{|\mathcal{P}_{t+1}|} \sum_{\mathbf{y} \in \mathcal{P}_{t+1}} \min_{\mathbf{x} \in \mathcal{P}_t} \|\mathbf{x} - \mathbf{y}\|_2^2. \qquad (6)$$

To evaluate the 3D structural fidelity, we calculate the F-score of the reconstructed point clouds. Specifically, the reconstructed point set $\mathcal{R}$ is obtained by back-projecting the predicted video depth into 3D space utilizing the intermediate camera poses predicted by our GEM module. Let $\mathcal{G}$ denote the corresponding ground-truth point set. Based on a specified distance threshold $d$, we define the precision $P(d)$ and recall $R(d)$ of the reconstruction $\mathcal{R}$ with respect to $\mathcal{G}$ as:

$$P(d) = \frac{1}{|\mathcal{R}|} \sum_{\mathbf{r} \in \mathcal{R}} \mathbb{I}\big[\min_{\mathbf{g} \in \mathcal{G}} \|\mathbf{r} - \mathbf{g}\|_2 < d\big], \quad R(d) = \frac{1}{|\mathcal{G}|} \sum_{\mathbf{g} \in \mathcal{G}} \mathbb{I}\big[\min_{\mathbf{r} \in \mathcal{R}} \|\mathbf{g} - \mathbf{r}\|_2 < d\big], \qquad (7)$$

where $\mathbb{I}[\cdot]$ denotes the Iverson bracket, which equals $1$ if the condition is true and $0$ otherwise. Precision measures the accuracy by calculating the percentage of reconstructed points that lie within the threshold $d$ from the ground truth, while recall measures completeness. To jointly capture both metrics, we report the F1-score, computed as their harmonic mean:

$$F_1(d) = \frac{2 \cdot P(d) \cdot R(d)}{P(d) + R(d)}. \qquad (8)$$

## B. Additional Experiment Results

### B.1. Sensitivity analysis of GemDepth to pose integration probability

As illustrated in Tab. 9, while GemDepth exhibits suboptimal performance in the complete absence of pose embeddings (0% probability), it achieves a significant performance leap once the poses predicted by GEM are introduced. Notably, the configuration with 50% integration probability yields results comparable to those of the 100% integration setting across all benchmarks. This trend demonstrates that GemDepth does not overfit to perfect geometric priors; instead, it effectively tolerates pose estimation noise. Consequently, our framework achieves robust video depth estimation even when facing imperfect geometric guidance from the GEM module.

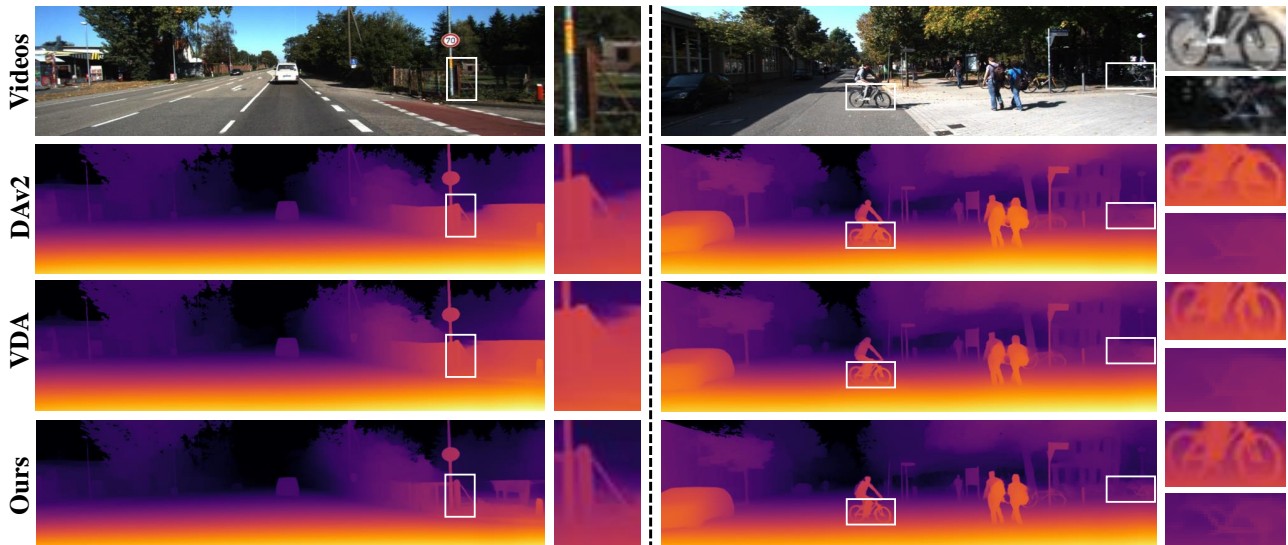

*Figure 9.* **More qualitative comparison on the KITTI (Geiger et al., 2013) dataset.**

## B.2. Quantitative Results on Videos of Varying Lengths

We evaluate the robustness of our model across varying video lengths using the Bonn (Palazzolo et al., 2019) dataset as a benchmark. Specifically, we conduct a comparative analysis between two versions of our GemDepth and the current state-of-the-art method, VideoDepthAnything (VDA) (Chen et al., 2025), under five different sequence lengths: 500, 400, 300, 200, and 100 frames.

As demonstrated in Tab. 10, both versions of GemDepth consistently outperform VDA across all tested durations. Our GemDepth-DAv2 achieves a significant average performance improvement of 25.4% over VDA. Furthermore, the GemDepth-VDA variant further pushes the boundary, delivering an impressive 29.5% average gain compared to the SOTA baseline. These results highlight that our models are not only highly effective in handling diverse temporal scales but also maintain a substantial performance edge regardless of the input video length.

## B.3. Comparison of Computational Complexity

As summarized in Table 11, we evaluate the computational efficiency of GemDepth in comparison to both state-of-the-art generative and discriminative models. GemDepth exhibits exceptional efficiency, operating at merely 12.6% of the FLOPs required by massive generative frameworks such as RollingDepth (Ke et al., 2025). Furthermore, when compared to the discriminative baseline VDA (Chen et al., 2025), our GemDepth achieves substantial improvements in 3D temporal consistency with only a marginal computational overhead, specifically an increase of 40G FLOPs and 10ms in latency. These results underscore that GemDepth maintains a superior balance between high-fidelity depth estimation and practical inference efficiency.

*Table 11.* Comparison of Computational Complexity

| Method | Latency(ms) | Params(M) | Flops(G) |
|---|---|---|---|
| RollingDepth | 240 | 1288 | 5382 |
| VDA | 85 | 405 | 640 |
| GemDepth | 95 | 584 | 680 |

## B.4. More Qualitative Comparisons

As illustrated in Fig. 9, we present a qualitative comparison between GemDepth and existing state-of-the-art methods, specifically DepthAnythingV2 (Yang et al., 2024b) and VideoDepthAnything (Chen et al., 2025), on the KITTI (Geiger et al.,

2013) dataset. By enlarging specific local regions, the superiority of our model in recovering fine-grained details and sharp contours is clearly demonstrated. While DepthAnythingV2 and VideoDepthAnything often misinterpret distant shadows as solid surfaces and exhibit blurring around complex boundaries like pillars or bicycles, GemDepth produces significantly sharper and more accurate results. These improvements stem from the ASTT module, which captures fine-grained spatial edges, and the GEM module, which enforces rigorous inter-frame geometric alignment. Consequently, our approach enhances edge sharpness while maintaining temporal consistency and overcoming occlusions through precise geometric reasoning.

