# OpenReview forum: "GemDepth: Geometry-Embedded Features for 3D-Consistent Video Depth"
_ICML.cc/2026/Conference — ICML 2026 regular_

### Official Review · Reviewer_18kX · 2026-03-05

**Soundness:** 3
**Presentation:** 3
**Significance:** 3
**Originality:** 3
**Overall Recommendation:** 4
**Confidence:** 3

**Summary:**

In this paper, the authors propose a framework, GemDepth, to maintain the 3D geometric consistency in depth estimation, which is based on injecting the information from camera motion and global 3D structure. The key contribution of the paper includes two modules: Geometry-Embedding Module (GEM) and Alternating Spatio-Temporal Transformer (ASTT). The GEM incorporates camera motion to improve intrinsic 3D perception and alignment capabilities. The ASTT resolves the conflict between detail preservation and temporal stability. With introducing these modules, the framework GemDepth show better depth estimation results in widely used datasets.

**Compliance With Llm Reviewing Policy:**

Affirmed.

**Final Justification:**

I would like to thank the authors for the detailed and comprehensive rebuttal. I appreciate the effort put into addressing my concerns, particularly in running the new evaluations. I would like to keep my original rating.

**Key Questions For Authors:**

Please refer to the weakness part.

**Limitations:**

No. The performance of the framework may degrade under significant illumination changes within the video.

**Strengths And Weaknesses:**

Strengths:
1. The paper is well organized and easy to follow.
2. The idea to injecting the camera motion into the framework for depth estimation is reasonable, which could provide more information to estimate depth with strong geometric consistency.
3. The ablation studies and experiments in representative benchmarks show the effectiveness of proposes modules.
4. A two stage training is a meaningful strategy to mitigate the bottleneck in data with ground truth camera poses.

Weaknesses:
1. This framework incorporates camera motion for more geometric consistency in depth estimation; however, it would be better to show a results for camera estimation, since the ground truth camera poses also were used in training the framework.
2. DepthAnything v3 still show the results that can estimate the consistent depth and corresponding camera poses. It would be better to show a comparison in depth estimation, camera pose estimation, point cloud visualization with DepthAnything v3.
3. I would like to know the difference between this framework and VGGT. I hope that the authors can illustrate it with more details.

---

> ### Author Rebuttal · Authors · 2026-03-30
>
> We sincerely thank Reviewer 18kX for recognizing our well-organized presentation, the effectiveness of our geometry-guided design, and our two-stage training strategy. Below, we address your constructive feedback regarding baseline comparisons (Depthanything3, VGGT), camera pose estimation and differences between GemDepth and VGGT.
>
> **Q1:Comparison with DepthAnything3 (DA3) and Camera Pose Estimation.**
>
> We completely agree that evaluating camera pose estimation and comparing it with state-of-the-art models like DepthAnything3 is crucial, as both frameworks jointly consider depth and pose.
>
> We compared GemDepth with Depthanything3 and VGGT in terms of video depth accuracy, video depth temporal consistency, video pose prediction accuracy (ATE) and point cloud estimation (CD). For depth estimation, we strictly follow the data organization and testing methodology established in VideoDepthAnything (VDA) to ensure a fair comparison. Compared with Depthanything3, GemDepth uses only ~50% parameters yet slashes the depth metrics AbsRel by 47% and TAE by 58% (Scannet). Concurrently, maintaining highly comparable 3D metrics (ATE, CD) to massive foundation models demonstrates the unique superiority of our algorithm. This proves our efficient geometry-guided design successfully enforces 3D structural integrity without the massive computational overhead of foundation models .
>
> **Table 1: Quantitative comparison on Scannet and Bonn datasets.** AbsRel/TAE evaluate 2D depth; ATE (trajectory error) and CD (Chamfer Distance) measure 3D structural fidelity.
> | Method | Params  | AbsRel (Scannet 2D) | TAE (Scannet 2D) | ATE (Scannet 3D) | CD (Scannet 3D) | AbsRel (Bonn 2D) | TAE (Bonn 2D) | ATE (Bonn 3D) | CD (Bonn 3D) |
> | :--- | :---: | :---: | :---: | :---: | :---: | :---: | :---: | :---: | :---: |
> | DepthAnything3 | 1.10B | 0.125 | 1.12 | **0.021** | **10.42** | 0.198 | 2.38 | **0.0098** | 1.81 |
> | VGGT | 1.19B | 0.132 | 1.92 | 0.028 | 10.45 | 0.184 | 2.87 | 0.014 | **1.73** |
> | **GemDepth (Ours)**| **0.58B** | **0.066** | **0.47** | 0.025 | 10.49 | **0.051** | **2.03** | 0.013 | 2.39 |
>
> **Q2:Fundamental Differences between GemDepth and VGGT.**
>
> We appreciate the opportunity to clarify our fundamental differences from VGGT. While both address video depth, they diverge completely in paradigm, mechanisms, and accuracy:
>
> 1. Paradigm & Task Objective: VGGT is a general-purpose, multi-task foundation model designed to output multiple 3D parameters concurrently. In contrast, GemDepth is a dedicated framework explicitly tailored to solve the long-standing "2D-3D gap" in video depth. We pioneer a "pose-first" paradigm designed specifically to eradicate temporal flickering, demanding much higher precision in depth metrics.
>
> 2. The Functional Role of Pose: In VGGT, the camera pose is treated merely as a parallel output target. In GemDepth, the predicted pose serves a fundamentally different purpose: it acts as an explicit physical prior injected back into the network. Guided by these geometric embeddings, our unique goal is to explicitly enforce latent point-level correspondences across dynamic 2D frames, definitively resolving the inherent conflict between spatial sharpness and temporal stability.
>
> 3. Spatiotemporal Mechanism: VGGT relies on flattened global-local attention, which inherently dilutes dense inter-frame temporal dynamics. Conversely, to achieve the aforementioned correspondences, our Alternating Spatio-Temporal Transformer (ASTT) introduces a dedicated temporal attention mechanism. Strictly conditioned on the injected geometric priors, ASTT rigorously preserves 3D structural integrity rather than flattening it.
>
> 4. Empirical Superiority (Table 1): These architectural distinctions translate into massive performance gaps. As shown in Table 1, while maintaining highly comparable 3D metrics (ATE, CD) to VGGT, GemDepth slashes depth errors (reducing AbsRel by 50.0% and TAE by 75.5%) despite utilizing only ~50% of the parameters. This definitively proves the algorithmic superiority of our approach for video depth.
>
> **Q3:Discussion on Limitations(Illumination Changes)**
>
> We thank the reviewer for this practical insight and have added it to our Limitations section. Significant lighting changes violate the photometric constancy assumption, hindering the GEM module's inter-frame feature matching. Consequently, while the spatial branch still yields reliable single-frame depth, the weakened geometric prior can degrade strict temporal consistency and cause localized flickering.

---

> > ### Author Rebuttal · Reviewer_18kX · 2026-04-03
> >
> > I would like to thank the authors for the detailed and comprehensive rebuttal. I appreciate the effort put into addressing my concerns, particularly in running the new evaluations. I would like to keep my original rating at this stage.

---

> > > ### Author Response · Authors · 2026-04-03
> > >
> > > Thanks for your acknowledge of our work. Thanks again for your time and effort in reviewing our paper.

---

### Official Review · Reviewer_MrpS · 2026-03-12

**Soundness:** 3
**Presentation:** 4
**Significance:** 4
**Originality:** 3
**Overall Recommendation:** 4
**Confidence:** 3

**Summary:**

The author propose a generalized framework for video depth estimation, named GemDepth, with a Geometry-Embedding Module (GEM) that injects pose-aware embeddings, enforcing 3D structural constraints and consistency within video depth estimation. The authors introduce a ASTT module which leverages geometric priors to decouple dependency modeling. This work adopts a two-stage training strategy.
Their approach claims to need minimal training samples to beat SOTA.

**Compliance With Llm Reviewing Policy:**

Affirmed.

**Final Justification:**

The authors have addressed most of my concerns within their rebuttal. As such, I shall keep my current rating.

**Key Questions For Authors:**

Key Questions for Authors

1. For the loss denoted as L_{total}. Have the authors ablated the loss coefficients α, β and γ. What justification is there for the use of these coefficients given their current assigned values?

2.1 For inter-frame stability / temporal consistency (see Table 2), the authors have only tracked one metric (TAE↓). The authors may consider adding additional temporal metrics (e.g., Temporal PSNR) to evaluate the performance of the proposed method in video/temporal settings.

2.2 The author may look to show additional dataset analysis (see Table 2), to strengthen claims on best temporal consistency. An additional dataset here will be useful to show generalisation of the proposed approach.

3.0 Whilst runtime is reported in Table 1, the authors may consider reporting compute performance VRAM usage and GFLOPS of the proposed framework against SOTA. This can better raise compute against performace analysis.

4.0 The authors state the training time until convergence. They may look to report which stopping criterion they used as well as the full training epoch length for the experiment.

**Limitations:**

No: Limitations of the proposed technique have not been discussed. The authors should look to include limitations of their work. For example; how does their method work in long-term scenes (long length videos with many scene changes) - this is not discussed at large within the proposed work.

Yes: Societal impact of their work has been discussed (although to a general high-level; discussion around the area as a whole).

**Strengths And Weaknesses:**

The authors propose a novel design, and highlight SOTA performance in the results. The author have included thorough ablation of their proposed modules collective contributions.

Soundness
The claims raised in this work are supported by quantitative and qualitative results.
The authors propose through results with multiple datasets. However, the authors may consider including more depth metrics to deepen the qualitative results (e.g., RMSE) to strengthen their claims across more metrics.

Presentation
This paper is presented well. There is no major presentation issues to comment on.
Pipeline figure is showcased well.

Significance
Video Depth Estimation is important for many downstream tasks in deep learning/computer vision community.
This paper aims to address (i) fine-detail , and (ii) temporal inconsistencies within Video Depth Estimation. (i) fine-detail is explicitly raised through the limitations of existing SOTA work (see Figure 1).
The authors have shown that irrespective of the instantiation used (DepthAnything-V2 or VideoDepthAnything), that their approach improves on the reported metrics Absrel↓ and δ1↑.

Originality
This works design is original in design with 2 novel modules GEM and ASTT. The GEM is build upon a lightweight EfficientPoseNet (Wang et al., 2024), and the ASTT utilises RoPE (Su et al., 2024) and inter-frame Idx Embedding (Yang et al., 2025). However, both modules have insight driven designs that are original and are built upon/integrate the previous methods.
The GEM enforces 3D structure (via pose-aware embeddings), and the ASTT module which leverages geometric priors to decouple dependency modeling.

---

> ### Author Rebuttal · Authors · 2026-03-30
>
> We sincerely thank Reviewer MrpS for the constructive suggestions regarding additional metrics, compute analysis, and limitations.
>
> **Q1:Ablation Study on Loss Weights.**
>
> Our coefficients are determined through established practices and empirical tuning:
>
> $\alpha=0.5$ and $\beta=10$: To ensure stable training and fair baseline comparisons, we strictly follow the proven configuration of VideoDepthAnything (VDA). These values optimally balance spatial and temporal gradients, obviating the need for redundant ablation.
>
> $\gamma$ (Camera loss): For our novel $\mathcal{L}_{cam}$, we empirically ablated $\gamma$ to prevent the pose supervision from overwhelming the primary depth task. The optimal balance occurs when it contributes ~20% to the total loss. Higher proportions compromise 2D depth accuracy, while lower ones weaken 3D structural constraints.
>
> **Q2:Additional Temporal Metrics.**
>
> We compared the 2D temporal stability and 3D structural consistency of GemDepth against the state-of-the-art baseline, VideoDepthAnything (VDA). Here, 3D consistency is evaluated using the temporal Chamfer Distance (CD). Specifically, we project the predicted video depth maps into 3D point clouds using ground-truth (GT) camera poses, and calculate the CD between the aligned point clouds of consecutive frames. This allows us to rigorously isolate and measure the pure temporal structural fidelity of the depth itself. Our method achieves significant improvements across both evaluation dimensions. Most notably, the temporal Chamfer Distance (CD) errors are reduced by 10.9% and 40% across the two datasets, compellingly validating the inherent 3D-aware capabilities of our architecture.
>
> | Method | CD (Scannet 3D) | TAE (Scannet 2D) | CD (Bonn 3D) | TAE (Bonn 2D) |
> | :--- | :---: | :---: | :---: | :---: |
> | VDA | 0.92 | 0.570 | 0.035 | 2.11 |
> | GemDepth (Ours) | **0.82** | **0.471** | **0.021** | **2.03** |
>
> **Q3:Additional dataset analysis.**
>
> We strongly agree that evaluating on diverse datasets is crucial to demonstrating the generalization of our temporally consistent design. To strengthen our claims, we have expanded our temporal consistency analysis (Table 2 in the main paper) to include a comprehensive suite of four diverse datasets: Sintel, Bonn, KITTI, and Scannet.
>
> As shown in the expanded table below, GemDepth consistently achieves a lower Temporal Alignment Error (TAE) compared to the state-of-the-art VideoDepthAnything (VDA) across all four datasets, yielding an average reduction of 11.7%. This robust performance across varied dynamic scenes and camera motions compellingly validates the strong generalization capabilities of our proposed architecture.
>
> | Method   | Sintel | Bonn | Kitti | Scannet |
> | :--- | :---: | :---: | :---: | :---: |
> | VDA | 0.538 | 2.11 | 0.703 | 0.570 |
> | GemDepth (Ours) | **0.524** | **2.03** | **0.542** | **0.471** |
>
> **Q4:Comparison of Computational Complexity.**
>
> We strongly agree that analyzing computational overhead is essential. As shown below, we benchmarked the latency, parameters, and FLOPs of our framework against state-of-the-art baselines.
>
> | Method | Latency(ms) | Params(M) | Flops(G) |
> | :--- | :---: | :---: | :---: |
> | RollingDepth | 240 | 1288 | 5382 |
> | VDA | 85 | 405 | 640 |
> | GemDepth | 95 | 584 | 680 |
>
> The compute-to-performance trade-off of GemDepth is highly favorable. Compared to massive generative models like RollingDepth, GemDepth operates with merely 12.6% of the FLOPs and significantly lower latency. Furthermore, compared to our direct baseline (VDA), our geometry-aware modules introduce a marginal overhead of just +40G FLOPs and +10ms latency. This minimal cost strongly justifies our design, given the substantial gains in 3D temporal consistency.
>
> **Q5:Training Details and Stopping Criterion.**
>
> We apologize for omitting these details and will add them to Section 4.1.
>
> **Epoch Length**: Stage 1 was trained for 130k steps(approximately 2 epoch). Stage 2  was trained for 144k steps(approximately 1.5 epoch). Throughout both stages, each training iteration processes a continuous sequence of 32 video frames as input.
>
> **Stopping Criterion:** We employ a rigorous Early Stopping strategy based on a hold-out validation set. Specifically, we evaluate the model metrics every 5,000 steps. Training is halted if the validation error does not show a significant decrease over a continuous span of 20,000 steps. This ensures the model fully converges without overfitting.
>
> **Q6:Discussion on Limitations(Long-term Scenes and Scene Changes).**
>
> We appreciate this insightful feedback and have added a dedicated Limitations section. Abrupt scene changes inherently break the visual continuum required by the GEM module, disrupting cross-frame feature matching. Fortunately, our sliding-window strategy automatically re-establishes a geometric anchor for new scenes, successfully preventing accumulated trajectory drift and global failure.

---

> > ### Author Rebuttal · Reviewer_MrpS · 2026-04-03
> >
> > Thank you for the rebuttal, and for providing these further evaluations. I have read through the authors rebuttal carefully, and they have addressed each of my previous concerns. As such, I shall keep my current score.

---

> > > ### Author Response · Authors · 2026-04-04
> > >
> > > Thanks for your acknowledge of our work. Thanks again for your time and effort in reviewing our paper.

---

### Official Review · Reviewer_hBz4 · 2026-03-13

**Soundness:** 2
**Presentation:** 2
**Significance:** 2
**Originality:** 2
**Overall Recommendation:** 3
**Confidence:** 3

**Summary:**

The paper introduces GemDepth, a framework designed to improve the spatial accuracy and temporal consistency of video depth estimation. The authors argue that existing discriminative models lack explicit 3D geometric awareness. To address this, they propose two main components: the Geometry-Embedding Module (GEM), which predicts inter-frame 6-DoF camera poses and a global scale factor to create explicit motion priors , and the Alternating Spatio-Temporal Transformer (ASTT), which uses these priors to decouple dependency modeling into temporal geometric alignment and spatial structural refinement. Furthermore, the authors utilize a two-stage training strategy to overcome the scarcity of camera-pose annotated data, optimizing the geometry components first before freezing them to fine-tune the rest of the network. Built upon baselines like DepthAnything V2 and VideoDepthAnything, GemDepth demonstrates state-of-the-art zero-shot performance across multiple benchmarks.

**Compliance With Llm Reviewing Policy:**

Affirmed.

**Final Justification:**

I appreciate the authors' response; however, I remain concerned about the proposed method's performance on dynamic scenes, particularly given its inferior 3D metrics compared to generic models like VGGT. Looking ahead, I am unconvinced that the intermediate pose representation can ultimately outperform these generic models on general videos. Specifically, it remains unclear how relying on camera pose provides any advantage in scenarios featuring static cameras and dynamic scenes.

**Key Questions For Authors:**

- **Handling Dynamic Objects**: The GEM module explicitly models global 6-DoF camera ego-motion. However, in Section 4.3, it is claimed that this global structural consistency helps reconstruct independent dynamic objects, such as airborne balloons. Could the authors clarify the mechanism by which a rigid, global camera pose prior resolves independent, non-rigid dynamic object motion? Is there a risk that the temporal attention mechanism heavily penalizes dynamic objects whose trajectories conflict with the global ego-motion prior?

- **Catastrophic Pose Failure**: The sensitivity analysis in Appendix B.1 demonstrates robustness when pose embeddings are dropped out with a certain probability. However, how does the framework behave when the GEM module actively predicts a *corrupted or entirely incorrect* pose (e.g., during extreme out-of-distribution movements or severe occlusions)? Does the sequential reliance on GEM cause a complete degradation of the depth prediction?

- **Scale Consistency Between Pose and Depth**: In Section 3.2, the framework addresses monocular scale ambiguity in camera ego-motion by computing a global scale factor $Z$ and normalizing translations before encoding them. However, in Section 3.4, the depth prediction is supervised using a scale- and shift-invariant loss ($\mathcal{L}_{ssi}$), which inherently ignores the absolute metric scale of the depth map. Given that the pose and depth are operating in seemingly decoupled scale spaces during training, how does GemDepth ensure that the implicit scale of the predicted depth map is consistent with the scale of the predicted camera translation? If the scales are mismatched, wouldn't the geometric constraints injected by the GEM module operate in a distorted metric space?

- **Lack of Quantitative 3D Metrics**: The paper's title and core motivation heavily emphasize achieving "3D-Consistent Video Depth," and the qualitative results (Figures 3 and 8) showcase fused point clouds to demonstrate this structural integrity. However, the quantitative evaluation relies entirely on 2D spatial metrics (AbsRel, $\delta_1$) and a 2D reprojection metric (TAE). Given that the framework explicitly models 6-DoF camera poses and generates these point clouds, could the authors also report the standard 3D reconstruction metrics (e.g., Chamfer Distance or F-Score after multi-view fusion)? Including these metrics would provide a much more rigorous and convincing proof of the claimed 3D geometric consistency than visual inspection alone.

**Limitations:**

No. The authors have not adequately discussed the limitations and potential negative societal impact of their work. They provide a brief "Impact Statement" focused purely on the positive societal consequences of robust machine perception.

The authors are encouraged to include a dedicated "Limitations" section. This section should transparently discuss the boundaries of their approach, for example:

- The sequential bottleneck created by relying on the GEM module's pose accuracy.

- The potential conflicts between global ego-motion priors and highly dynamic, independent object motion.

**Strengths And Weaknesses:**

**Soundness**

- **Strengths**: The experimental methodology is rigorous, evaluating the framework on multiple diverse datasets (KITTI, Sintel, Scannet, Bonn) using standard spatial metrics (AbsRel, $\delta_{1}$) and temporal metrics (Temporal Alignment Error). The claims regarding the utility of individual modules are well-supported by extensive ablation studies.

- **Weaknesses**: The architecture relies on a strictly sequential dependency; depth prediction is heavily conditioned on the camera pose prediction generated by the GEM module. While the authors demonstrate that the ASTT module can tolerate simulated pose noise via a probability dropout, it is methodologically unclear how the network behaves if the GEM module catastrophically fails (e.g., predicting wildly incorrect poses due to extreme motion blur or occlusion). Additionally, the GEM module predicts a single global 6-DoF camera pose. Predicting global rigid ego-motion is theoretically insufficient for modeling independent, non-rigid dynamic objects, casting doubt on the soundness of the claim that global structural consistency explicitly preserves the geometric integrity of highly dynamic subjects.


**Presentation**

- **Strengths**: The submission is clearly written, logically structured, and easy to follow. The authors effectively contrast their approach against both generative and discriminative baselines. The inclusion of architectural diagrams (Figure 4) and visual decompositions (Figure 5) significantly aids in understanding the proposed ASTT and GEM modules.

- **Weaknesses**: The presentation omits a dedicated discussion of the framework's limitations and specific failure modes within the main text. In addition, I think it's uncommon to refer to figures in the abstract, and it might be better to move the reference to the introduction section.

**Significance**

-  **Strengths**: The paper addresses a highly relevant problem, as temporally stable video depth estimation is a critical prerequisite for downstream tasks like autonomous driving and augmented reality. The framework proves its practical utility by achieving significant performance gains with only a marginal computational overhead (less than 20% increase in inference latency over baselines).

- **Weaknesses**: I am concerned that relying on camera pose estimation for depth prediction might pose a constraint for the robustness of the depth prediction and the results on dynamic scenes.

**Originality**

- **Strengths**: The way the ASTT module explicitly decouples spatio-temporal modeling is different from previous methods that don't rely on camera pose.

- **Weaknesses**: The concept of using camera pose for depth estimation has been explored in previous works, for example, Consistent Video Depth Estimation (Luo et al., SIGGRAPH 2020), and DeepV2D: Video to Depth with Differentiable Structure from Motion (Teed & Deng, ICLR 2020). However, the implementation of this paper is different.

---

> ### Author Rebuttal · Authors · 2026-03-30
>
> **Q1:Handling Dynamic Objects.**
>
> We thank the reviewer for this insightful question. While a rigid global pose might theoretically penalize dynamic objects, GemDepth effectively avoids this through two specific architectural mechanisms:
>
> 1. Implicit Disentanglement via Soft Priors: The global pose predicted by GEM effectively captures the dominant static background, serving as a clear reference point. Crucially, GEM does not apply hard pixel-level warping. Instead, it transforms this background motion into implicit Geometric Embeddings. During training, ASTT utilizes these embeddings as soft priors: the self-attention mechanism adaptively down-weights the temporally inconsistent global features for dynamic pixels. This implicitly disentangles dynamic objects from the background without heavily penalizing them.
>
> 2. Spatial Refinement Compensation: The decoupled nature of ASTT plays a critical role. ASTT alternates Temporal and Spatial Attention. Even if Temporal Attention struggles to extract effective temporal features at dynamic regions due to global pose violations, the subsequent Spatial Attention—empowered by robust DINOv2 semantics—immediately compensates. It repairs high-frequency structural details within a single frame, seamlessly preserving crisp dynamic contours.
>
> Empirical Evidence (Bonn Dataset): This capability is definitively validated on Bonn's highly dynamic sequences. Quantitatively (Tab. 1 in the main paper), GemDepth achieves a massive 28.2% AbsRel improvement over VDA (0.051 vs. 0.071). Qualitatively (Fig. 7 in the main paper), it preserves sharp contours of fast-moving objects (e.g., airborne balloons), entirely avoiding the severe blurring caused by blind smoothing.
>
> **Q2:Catastrophic Pose Failure.**
>
> The sequential reliance on GEM does not lead to a complete degradation of depth prediction; instead, the network exhibits graceful degradation. To validate this, we conducted an extreme stress test by injecting up to 100% relative Gaussian noise (scaled to the original translation and quaternion magnitudes) into the pose predictions of the GEM module.
>
> | Add Noise (Trans & Quat) | 0% | 5% | 10% | 20% | 50% | 100% |
> | :--- | :---: | :---: | :---: | :---: | :---: | :---: |
> | AbsRel | 0.066 | 0.066 | 0.067 | 0.071 | 0.078 | 0.096 |
> | TAE  | 0.472 | 0.475 | 0.480 | 0.503 | 0.577 | 0.638 |
>
> As shown, even when the pose is catastrophically corrupted, the depth prediction merely degrades. The network learns to naturally suppress conflicting geometric tokens and rely heavier on spatial appearance features.
>
> **Q3:Scale Consistency Between Pose and Depth.**
>
> While our framework operates without an absolute metric scale, it strictly avoids metric distortion through our specific training design:
> 1. Normalized Pose Supervision: As our fundamental task is relative video depth estimation, the $\mathcal{L}_{ssi}$ loss is essential. To align with this relative nature, the GEM module is explicitly supervised using scale-normalized ground-truth translations. Thus, the predicted poses inherently operate within a unified, normalized scale space from the very beginning, eliminating any initial scale inconsistency.
> 2. Scale Alignment via Conditioning: During the ASTT fine-tuning stage, depth generation is strictly conditioned on these normalized pose embeddings. Crucially, because ASTT uses these poses to establish temporal geometric relationships, the network must align the implicit scale of the depth maps with the normalized pose scale to minimize the temporal geometric matching loss ($\mathcal{L}_{tgm}$).
>
> **Q4:Lack of Quantitative 3D Metrics:**
>
> To address this, we projected our predicted depth sequences into 3D point clouds using GEM poses and evaluated them on the Scannet and Bonn datasets. For depth estimation, we strictly follow the data organization and testing methodology established in VideoDepthAnything (VDA) to ensure a fair comparison. As shown below, we benchmarked against massive foundation models (DepthAnything3, VGGT) across 2D (AbsRel, TAE) and 3D (ATE, Chamfer Distance) metrics.
>
> GemDepth significantly outperforms these baselines in 2D metrics while achieving highly competitive 3D structural fidelity. Crucially, our model delivers this comparable 3D quality with a mere fraction of their parameters.
>
> Furthermore, in the field of video depth estimation, our 3D metrics consistently surpass the SOTA method VDA. Please refer to our response to Reviewer MrpS (Q2).
>
> | Method | Params  | AbsRel (Scannet) | TAE (Scannet) | ATE (Scannet) | CD (Scannet) | AbsRel (Bonn) | TAE (Bonn) | ATE (Bonn) | CD (Bonn) |
> | :--- | :---: | :---: | :---: | :---: | :---: | :---: | :---: | :---: | :---: |
> | DepthAnything3 | 1.10B | 0.125 | 1.12 | **0.021** | **10.42** | 0.198 | 2.38 | **0.0098** | 1.81 |
> | VGGT | 1.19B | 0.132 | 1.92 | 0.028 | 10.45 | 0.184 | 2.87 | 0.014 | **1.73** |
> | **GemDepth (Ours)**| **0.58B** | **0.066** | **0.47** | 0.025 | 10.49 | **0.051** | **2.03** | 0.013 | 2.39 |

---

> > ### Author Rebuttal · Reviewer_hBz4 · 2026-04-03
> >
> > I appreciate the authors' response. After checking the new results, I have significant concerns about whether the proposed model can work well on dynamic scenes. In particular, the authors presented an additional evaluation of Chamfer Distance (CD), which I have copy-pasted below:
> >
> > | Method              | CD (Scannet) | CD (Bonn) |
> > | :------------------ | :----------- | :-------- |
> > | DepthAnything3      | **10.42**    | 1.81      |
> > | VGGT                | 10.45        | **1.73**  |
> > | **GemDepth (Ours)** | 10.49        | 2.39      |
> >
> > Note that ScanNet is static, while Bonn is dynamic. These results seem to confirm my initial concern that the proposed intermediate pose representation might work well for static scenes, but not for dynamic ones. Here, we can see a clear performance gap on the dynamic Bonn dataset when compared to generic models like DA3 and VGGT. I am not sure if the authors have explanations to the clearly worse results on the dynamic Bonn dataset.

---

> > > ### Author Response · Authors · 2026-04-03
> > >
> > > **Response to the performance on dynamic scenes and the scope of our task**
> > >
> > > We sincerely thank the reviewer for the rigorous follow-up and the sharp observation regarding the Chamfer Distance (CD) on the dynamic Bonn dataset. However, we respectfully disagree with the conclusion that our method fails in dynamic scenes. We clarify this from three critical perspectives:
> > >
> > > 1. Clarifying the Task Boundary
> > > Our primary objective is discriminative Video Depth Estimation, not serving as a universal 3D geometry foundation model. Generic models like VGGT/DA3 are trained on massive datasets (exceeding our training data by over 10 times) specifically to output robust global 3D geometry. In highly dynamic scenes (like Bonn), moving objects naturally introduce noise into our intermediate pose representation. Crucially, our module leverages this noisy pose merely as a soft prior to improve depth —our true objective(the highest depth accuracy among all evaluated sota methods). Since CD evaluates globally projected point clouds, a perturbed global pose inevitably degrades CD, regardless of the underlying depth's precision.
> > >
> > > 1. Robust Performance in Highly Dynamic Scenes (KITTI & Sintel & MVS_Synth)
> > > To definitively dispel concerns regarding dynamic environments, we introduce new evaluations on the highly dynamic KITTI dataset (Sequence 02, featuring fast-moving vehicles), Sintel dataset and MVS_Synth dataset. As shown in Table 1, GemDepth outperforms DA3 across both benchmarks. Specifically, on KITTI, our method achieves a staggering 43.2% reduction in AbsRel (0.125 %\rightarrow% 0.071) while maintaining a highly comparable 3D CD metric (25.96 vs. 26.88). This strong performance extends to the Sintel dataset, where GemDepth delivers a remarkable 58.9% reduction in AbsRel (0.382 $\rightarrow$ 0.157) and improves the 3D CD metric from 9.79 to 9.01. Similarly, on the MVS-Synth dataset, we observe a 56.4% decrease in AbsRel (0.094 $\rightarrow$ 0.041) alongside an enhanced 3D CD score (20.41 $\rightarrow$ 18.56). This clearly demonstrates that our method does not fail in highly dynamic scenes.
> > >
> > > 1. Fair Comparison within the Video Depth Domain
> > > A true apples-to-apples comparison must be conducted against peer Video Depth Estimation models. (Metric Clarification: Unlike Table 1, the CD in Table 2 is computed using Ground-Truth (GT) poses, because generic video depth models cannot predict camera poses. This directly measures pure 3D depth consistency). As demonstrated in Table 2, when evaluated on a level playing field, GemDepth completely dominates all state-of-the-art video depth peers. Particularly for 3D CD, we outperform the best peer (VDA) by 10.9% on MVS_Synth (0.587 $\rightarrow$ 0.523), 22.5% on Sintel (0.102 $\rightarrow$ 0.079), 34.4% on KITTI (0.384 $\rightarrow$ 0.252) and 40.0% on Bonn (0.035 $\rightarrow$ 0.021).
> > >
> > > In summary, while we may not beat foundation models at their own global geometry prediction game, we show a clear advantage over existing methods in our true objective: extracting highly accurate and consistent video depth in dynamic scenes.
> > >
> > > **Table 1:Comparison with Generic Foundation Models on Highly Dynamic Scenes.**
> > > | Method/Metrics | MVS_Synth (AbsRel) | MVS_Synth (TAE) | MVS_Synth (CD) | Sintel (AbsRel) | Sintel (TAE) | Sintel (CD) | KITTI (AbsRel) | KITTI (TAE) | KITTI (CD) | Bonn (AbsRel) | Bonn (TAE) | Bonn (CD) |
> > > | :--- | :---: | :---: | :---: | :---: | :---: | :---: | :---: | :---: | :---: | :---: | :---: | :---: |
> > > | VGGT | 0.113 | 0.281| 21.69 | 0.551 | 0.529| 10.23 | 0.226 | 0.774 | 29.85 | 0.198 | 2.38 | 1.81 |
> > > | DA3 | 0.094 | 0.264 | 20.14 | 0.382 | 0.536 | 9.79 | 0.125 | 0.789 | 26.88 | 0.184 | 2.87 | **1.73** |
> > > | GemDepth | **0.041** | **0.255** | **18.56** |  **0.157** | **0.524** | **9.01** | **0.071** | **0.542** | **25.96** | **0.051** | **2.03** | 2.39 |
> > >
> > > **Table 2:Comparison with State-of-the-Art Video Depth Estimation Models.**
> > > | Method/Metrics | MVS_Synth (AbsRel) | MVS_Synth (TAE) | MVS_Synth (CD) |  Sintel (AbsRel) | Sintel (TAE) | Sintel (CD) | KITTI (AbsRel) | KITTI (TAE) | KITTI (CD) | Bonn (AbsRel) | Bonn (TAE) | Bonn (CD) |
> > > | :--- | :---: | :---: | :---: | :---: | :---: | :---: | :---: | :---: | :---: | :---: | :---: | :---: |
> > > | DepthCrafter | 0.088| 0.329 | 0.615 | 0.299| 0.577 | 0.158 | 0.164 | 0.754 | 0.398 | 0.153 | 2.35 | 0.038 |
> > > | RollingDepth | 0.085 | 0.364 | 0.599 | 0.417 | 0.589 | 0.166 | 0.107 | 0.737 | 0.455 | 0.088 | 2.46 | 0.042 |
> > > | DAv2 | 0.126 | 0.413 | 0.752 | 0.390 | 0.869 | 0.265 | 0.137 | 1.056 | 0.468 | 0.127 | 2.87 | 0.082 |
> > > | VDA | 0.069 | 0.258 | 0.587 | 0.295 | 0.538 | 0.102 | 0.083 | 0.703 | 0.384 | 0.071 | 2.11 | 0.035 |
> > > | GemDepth | **0.041** | **0.255** | **0.523** | **0.157** | **0.524** | **0.079** | **0.071** | **0.542** | **0.252** | **0.051** | **2.03** | **0.021** |

---

### Official Review · Reviewer_RkGs · 2026-03-13

**Soundness:** 3
**Presentation:** 2
**Significance:** 3
**Originality:** 2
**Overall Recommendation:** 3
**Confidence:** 4

**Summary:**

This paper proposes a new video depth estimation method with higher quality and better temporal consistency. The key contributions are two-fold: 1) a pose estimation module whose pose outputs are further encoded into camera tokens to improve motion awareness; 2) an additional transformer that handles temporal and spatial information by attending visual tokens from each image. Results show that the model outperforms DA2 and VDA, and ablation studies show the effectiveness of the proposed modules.

**Compliance With Llm Reviewing Policy:**

Affirmed.

**Final Justification:**

I thank the authors for the response. My concerns are partially addressed. However, the similarity issue between ASTT's spatial attention and VGGT's global attention remains unresolved, and the similarity concerns regarding Rayzer-like methods are not fully addressed. Considering these concerns, I will keep my original rating.

**Key Questions For Authors:**

Please see the Weakness section.

**Limitations:**

yes

**Strengths And Weaknesses:**

Strength
1. The model achieves notable improvements over the previous baselines.
2. The ablation study is relatively comprehensive and shows clear evidence of the proposed module's usefulness.

Weakness
1. Limited novelty. While I like the performance gain brought by the proposed approach, the technical contributions are relatively minor compared to previous approaches. For instance, using camera tokens as geometric cues is not novel in geometric foundation models overall.
    1. For instance, RayZer and its follow-up work, E-Rayzer, estimate camera poses and process them as tokens in a similar way that the proposed approach does. The difference is that RayZer uses the pose tokens as queries for novel view synthesis, while the paper focuses on depth quality. Though I fully understand the difference in task settings and the subtle design divergence, I still regarded them as similar in technical designs.
    2. The so-called Alternating Spatio-Temporal Transformer (ASTT) is essentially a self-attention with different tensor reshaping strategies. For tasks processing sequential data, this strategy is relatively straightforward to come up with. As a result, the proposed ASTT demonstrates similar design patterns as used in VGGT, which processes sequential data via per-frame attention first, followed up by global attention for all frames.


2. Lack of certain comparisons.
    1. Pose accuracy, and its correlation with temporal consistency and depth accuracy. The paper claims that the proposed GEM provides geometric embeddings that enforce consistency. Additional experiments on the quality of pose estimation and the correlation between pose accuracy and consistency/accuracy scores would be preferable to directly support the claim.
    2. Number of parameters, and FlOPs. While the paper provides the computation latency, it's not clear how many parameters and computations are involved in the new architectural design, which are crucial to evaluate the practical usages of the approach.

3. Minor Issues
    1. In the left image of Fig. 3, the trees on the left of the image seem to show a high degree of inconsistency, even for the proposed method. If it's the case, the actual temporal consistency of the method will be questioned. Some of the input images should be added to show the visual inputs in order to evaluate the fidelity of the reconstructed results.

---

> ### Author Rebuttal · Authors · 2026-03-30
>
> We deeply appreciate Reviewer RkGs for the positive and insightful feedback.
>
> **Q1:Limited novelty.**
> A1: While we share the formal mechanism of "early pose estimation" with RayZer and E-RayZer, this shared form does not diminish our fundamental novelty. Our core motivations and mechanisms are distinctly different:
>
> 1. Pioneering a Paradigm Shift: RayZer targets NVS by reconstructing static 3D scenes. In contrast, existing video depth models blindly rely on 2D temporal smoothing, entirely overlooking 3D physical constraints. GemDepth is the first to inject explicit geometry to bridge this critical "2D-3D gap" in discriminative video depth estimation. Therefore, our "early pose" approach is not merely a borrowed NVS trick, but a pioneering paradigm shift specifically designed to eradicate temporal flickering in the video depth domain.
>
> 2. Divergent Ultimate Goals: Both pipelines embed pose, but to entirely different ends. RayZer uses ray maps to condition a 3D scene builder. GemDepth does not build a 3D scene. Instead, our geometric embeddings serve exclusively as physical constraints to guide the ASTT module. Our goal is to explicitly enforce latent point-level correspondences across dynamic 2D frames , definitively resolving the inherent conflict between spatial sharpness and temporal stability.
>
> A2: We respectfully clarify that ASTT and VGGT employ different mechanisms for spatiotemporal interaction. VGGT's Frame (BT, L, C) and Global (B, TL, C) attentions lack pure temporal interaction, inherently diluting temporal dynamics. In contrast, ASTT introduces a dedicated Temporal Attention (BL, T, C). Coupled with pose embeddings, this forces interaction strictly across the time dimension for identical spatial locations, guaranteeing geometry-aware inter-frame consistency. This architectural advantage is strongly validated in Table 1, where our method significantly outperforms VGGT in video depth estimation, as measured by the AbsRel and TAE metrics.
>
> **Q2:Lack of certain comparisons.**
>
> Results of supplementary experiments are as follows:
>
> In Table 1, compared with DA3, GemDepth uses only ~50% parameters yet slashes the depth metrics AbsRel by 47% and TAE by 58% (Scannet). Concurrently, maintaining highly comparable 3D metrics (ATE, CD) to massive foundation models demonstrates the unique superiority of our algorithm.
>
> Table 2 evaluates robustness by injecting relative Gaussian noise (scaled to original translation and quaternion magnitudes) into the input poses. GemDepth maintains highly stable performance, showing gradual degradation only when noise exceeds 50%. This confirms our depth prediction is resilient and does not over-rely on perfect absolute poses.
>
> Table 3 highlights GemDepth's exceptional efficiency. It operates at just 12.6% of the FLOPs of massive models like RollingDepth. Compared to the VDA baseline, it achieves substantial 3D temporal consistency gains with only a marginal overhead (+40G FLOPs, +10ms latency).
>
>
> **Table 1: Quantitative comparison on Scannet and Bonn datasets.** AbsRel/TAE evaluate 2D depth; ATE (trajectory error) and CD (Chamfer Distance) measure 3D structural fidelity.
> | Method | Params | AbsRel (Scannet) | TAE (Scannet) | ATE (Scannet) | CD (Scannet) | AbsRel (Bonn) | TAE (Bonn) | ATE (Bonn) | CD (Bonn) |
> | :---: | :---: | :---: | :---: | :---: | :---: | :---: | :---: | :---: | :---: |
> | DepthAnything3 | 1.10B | 0.125 | 1.12 | **0.021** | **10.42** | 0.198 | 2.38 | **0.0098** | 1.81 |
> | VGGT | 1.19B | 0.132 | 1.92 | 0.028 | 10.45 | 0.184 | 2.87 | 0.014 | **1.73** |
> | **GemDepth (Ours)**| **0.58B** | **0.066** | **0.47** | 0.025 | 10.49 | **0.051** | **2.03** | 0.013 | 2.39 |
>
> **Table 2: Robustness evaluation under varying levels of injected pose noise on the Scannet dataset.**
> | Add Noise (Trans & Quat) | 0% | 5% | 10% | 20% | 50% | 100% |
> | :--- | :---: | :---: | :---: | :---: | :---: | :---: |
> | AbsRel | 0.066 | 0.066 | 0.067 | 0.071 | 0.078 | 0.096 |
> | TAE  | 0.472 | 0.475 | 0.480 | 0.503 | 0.577 | 0.638 |
>
> **Table 3: Comparison of Computational Complexity**
> | Method | Latency(ms) | Params(M) | Flops(G) |
> | :--- | :---: | :---: | :---: |
> | RollingDepth | 240 | 1288 | 5382 |
> | VDA | 85 | 405 | 640 |
> | GemDepth | 95 | 584 | 680 |
>
> **Q3:Minor issues of Figure 3**
>
> While acknowledging the minor background ghosting in Fig. 3, GemDepth preserves robust 3D structural fidelity across diverse scenes. We will include broader qualitative comparisons in the final revision. Crucially, our superior 3D temporal consistency is rigorously backed by quantitative metrics, as detailed in our response to Reviewer MrpS (Q2).
>
> In conclusion, GemDepth is the first to bridge the "2D-3D gap" in video depth estimation. Our experiments prove that compared to foundation models like VGGT, we achieve significant gains in 2D depth metrics while maintaining comparable 3D accuracy. We believe this work provides a highly meaningful contribution to the community.

---

> > ### Author Rebuttal · Reviewer_RkGs · 2026-04-01
> >
> > I want to thank the reviewers for the feedback. The response has addressed my concerns regarding computation efficiency and pose estimation. But the response to novelty issues is not fully convincing to me.
> >
> > 1. Regarding the use of pose tokens.
> >     1. While the authors emphasize the different task settings against Rayzer-like methods, both tasks essentially process the poses as tokens as intermediates in the framework (described as "early pose" in the paper).
> >     2. It also doesn't make sense to me when the authors claim that GemDepth "does not build a 3D scene". Depth estimation is essentially a 3D reconstruction task, with the major focus on 3D quality (as the 3D metrics requested by reviewer hBz4). The E-Rayzer paper serves the consistent goal with the additional capability of appearance modeling.
> >     3. Meanwhile, some potential overstatements seem to be misleading: "GemDepth is the _first to bridge the "2D-3D gap"_ in video depth estimation", "existing video depth models blindly rely on 2D temporal smoothing, _entirely overlooking 3D physical constraints_". These expressions could be revised to better reflect the contribution in a more precise and balanced manner.
> >
> > 2. The difference with VGGT.
> >     1. The authors state that "_VGGT's Frame (BT, L, C) and Global (B, TL, C) attentions lack pure temporal interaction, inherently diluting temporal dynamics_", which means that VGGT's global attention is conceptually the same as the proposed Spatial Attention, as processed in B × ( N × L ) × D in Figure 4 and Section 3.3.
> >     2. While I agree that the only difference lies in the temporal attention, which processes features as ( B × L ) × N × D, I found the performance advantage brought by this module is comparatively limited: In Table 3, removing the temporal attention from the ASTT (row 4 ---> row 1) leads to a minor performance decline (AbsRel, KITTI 0.094 -> 0.098, Sintel: 0.340 -> 0.342). I do not regard the DAv2 in Table 3 row 1 as a proper "Baseline", as it's a single-view model, which requires further basic processing (such as replacing the ASTT with a generic Transformer) to be a proper baseline.

---

> > > ### Author Response · Authors · 2026-04-02
> > >
> > > **Q1:Regarding the use of pose tokens.**
> > >
> > > We deeply appreciate the reviewer’s constructive feedback and sincerely apologize for any ambiguity in our previous phrasing. To clarify our distinct contributions:
> > >
> > > While both RayZer/E-RayZer and GemDepth utilize early pose tokens, their scene representations and optimization goals diverge significantly. E-RayZer optimizes for novel view rendering, requiring holistic 3D representations (geometry + appearance modeling). GemDepth, conversely, focuses exclusively on discriminative dense depth estimation.
> > >
> > > Consequently, our core novelty lies in the synergistic pipeline: not merely deriving pose tokens, but how these geometric priors fundamentally interact with visual features. E-RayZer uses pose tokens to condition a Gaussian/Latent scene reconstructor. In contrast, GemDepth introduces the ASTT module, which uniquely utilizes these tokens to physically constrain temporal attention, explicitly establishing latent point-level correspondences across frames.
> > >
> > > This is not merely a task switch, but a fundamental architectural innovation. As empirically proven in Table 1 of our previous response, while maintaining comparable holistic 3D metrics to comprehensive models like VGGT, our specific token-to-ASTT mechanism drastically slashes dense depth errors (reducing AbsRel by ~70% on Bonn). This performance leap effectively validates that our unique handling of pose tokens constitutes a substantial and novel contribution to the video depth domain. We will meticulously revise our previous claims for precision and add a dedicated discussion comparing these specific mechanisms with RayZer and VGGT.
> > >
> > > **Q2:The difference with VGGT.**
> > >
> > > We thank the reviewer for these deep insights. We agree that "Baseline + Spatial" essentially serves as a strong, generic spatio-temporal baseline akin to VGGT's global attention. However, we respectfully clarify the following two points regarding the performance gains and baseline selection:
> > >
> > > 1. Evaluating the True Value of Temporal Attention:The reviewer noted that adding Temporal Attention (Row 2 $\rightarrow$ Row 4) yields seemingly minor improvements in AbsRel. However, AbsRel is a single-frame evaluation metric, which inherently cannot reflect the module's primary objective: enforcing cross-frame 3D structural integrity. When evaluated on 3D temporal consistency metrics，the gains are highly significant. As shown in Table 1, even atop a strong spatio-temporal baseline, adding our ASTT module further reduces Scannet TAE by a notable 7.7% (0.775 to 0.715). This conclusively proves that our pure temporal attention explicitly resolves temporal flickering.
> > >
> > > 2. Rigorous Ablation on a Multi-View Baseline (VDA): We completely agree that comparing against a single-view baseline (DAv2) might not fully isolate our contribution. To demonstrate this more clearly, we present the detailed ablation results based on VideoDepthAnything (VDA). Since VDA serves as one of the primary multi-frame baselines in our main text, we had already conducted these comprehensive ablations during our initial experiments. VDA inherently provides strong temporal consistency priors.
> > > As demonstrated in Table 2, even on top of VDA's strong multi-view foundation, our temporal attention remains highly effective. Moving from the generic "Baseline + Spatial" (Row 2) to "Baseline + ASTT" (Row 4) slashes the TAE from 0.609 to 0.566 on Scannet (a notable 7.1% reduction). Furthermore, completing the Full Model with GEM priors yields the ultimate performance leap. This firmly demonstrates that our decoupled ASTT mechanism independently and substantially enhances temporal consistency over existing generic global attentions.
> > >
> > > **Table 1: Ablation study of proposed modules for DAv2 on Stage 1 (20k steps). Results on KITTI/Sintel (AbsRel), Scannet/Bonn (TAE & CD).**
> > > | Model | KITTI (AbsRel) | Sintel (AbsRel) | Scannet (TAE) | Scannet (CD) | Bonn (TAE) | Bonn (CD) |
> > > | :--- | :---: | :---: | :---: | :---: | :---: | :---: |
> > > | Baseline (DAv2) | 0.132 | 0.394 | 1.04 | 10.88 | 2.26 | 2.98 |
> > > | Baseline + Spatial | 0.098 | 0.342 | 0.775 | 10.75 | 2.18 | 2.72 |
> > > | Baseline + Temporal | 0.110 | 0.351 | 0.737 | 10.68 | 2.10 | 2.58 |
> > > | Baseline + ASTT | 0.094 | 0.340 | 0.715 | 10.65 | 2.09 | 2.53 |
> > > | Full Model | **0.088** | **0.328** | **0.654** | **10.61** | **2.06** | **2.46** |
> > >
> > > **Table 2: Ablation study for VDA.**
> > > | Model | KITTI (AbsRel) | Sintel (AbsRel) | Scannet (TAE) | Scannet (CD) | Bonn (TAE) | Bonn (CD) |
> > > | :--- | :---: | :---: | :---: | :---: | :---: | :---: |
> > > | Baseline (VDA) | 0.092 | 0.356 | 0.621 | 10.65 | 2.15 | 2.72 |
> > > | Baseline + Spatial | 0.082 | 0.337 | 0.609 | 10.62 | 2.12 | 2.68 |
> > > | Baseline + Temporal | 0.084 | 0.343 | 0.573 | 10.55 | 2.07 | 2.53 |
> > > | Baseline + ASTT | 0.080 | 0.328 | 0.566 | 10.52 | 2.05 | 2.47 |
> > > | Full Model | **0.077** | **0.308** | **0.550** | **10.49** | **2.03** | **2.38** ||

---

### Decision · Program_Chairs · 2026-04-30

**Decision:**

Accept (regular)

**Comment:**

The paper received mixed reviews, with several reviewers raising concerns about its novelty and performance gains in dynamic scenes. In their rebuttal, the authors provided extensive additional results on dynamic scenarios, along with more thorough comparisons against established baselines. Based on these clarifications, the meta-reviewers concluded that the manuscript constitutes a valuable contribution to the literature. The authors are requested to revise and improve the paper as per the discussion especially evaluation on dynamic scenarios.